

# Radar based high resolution ensemble precipitation analyses over the French Alps

Matthieu Vernay[1,2,3,4,5,6], Matthieu Lafaysse[1,2,3,4,5,6], and Clotilde Augros[5,2,4]

[1]Univ. Grenoble Alpes, 38400 Grenoble , France
[2]Université de Toulouse, 31100 Toulouse, France
[3]Météo-France
[4]CNRS
[5]CNRM
[6]Centre d'Études de la Neige, 38400 Grenoble , France

**Correspondence:** matthieu.vernay@meteo.fr

**Abstract.**

Reliable estimation of precipitation fields at high resolution is a key issue for snow cover modelling in mountainous areas, where the density of precipitation networks is far too low to capture their complex variability with topography. Adequate quantification of the remaining uncertainty in precipitation estimates is also necessary for further assimilation of complementary

snow observations in snow models. Radar observations provide spatialised estimates of precipitation with high spatial and temporal resolution, and are often combined with rain gauge observations to improve the accuracy of the estimate. However, radar measurements suffer from significant shortcomings in mountainous areas (in particular, unrealistic spatial patterns due to ground clutter). Precipitation fields simulated by high-resolution numerical weather prediction (NWP) models provide an alternative estimate, but suffer from systematic biases and positioning errors. Even though these uncertainties can be partially

described by ensemble NWP systems and systematic errors can be reduced by statistical post-processing, NWP precipitation estimates are still not reliable enough for the requirements of high resolution snow cover modelling.

In this study, better precipitation estimates are obtained through a specific analysis based on a combination of all these available products. First, a pre-processing step is proposed to mitigate the main deficiencies of radar and gauges precipitation estimation products, focusing on reducing unrealistic spatial patterns. This method also provides a spatialised estimate of the

associated error in mountainous areas, based on a climatological analysis of both radar and NWP-estimated precipitation. Three ensemble daily precipitation analysis methods are then proposed, first using only the modified precipitation estimates and associated errors, then combining them with ensemble NWP simulations based on the Particle Filter and Ensemble Kalman Filter data assimilation algorithms. The performance of the different precipitation analysis methods is evaluated at a local scale using independent ski resort precipitation observations. The evaluation of the pre-processing step shows its ability to remove

the main spatial artefacts coming from the radar measurements and to improve the precipitation estimates at the local scale. The local scale evaluations of the ensemble analyses do not demonstrate an additional benefit of ensemble NWP forecasts, but their contrasted spatial patterns are challenging to evaluate with the available data.



## 1 Introduction

Monitoring snow cover in mountainous areas is essential for a wide range of human activities and scientific applications (IPCC,
2022). The complex topography of these areas leads to a very high spatial variability of meteorological and snow conditions
(e.g. Clark et al., 2011), which is not fully sampled by any existing in situ observing network, especially at high altitudes
(Thornton et al., 2022). Operational applications such as water resource management or avalanche forecasting, which require
detailed monitoring of meteorological conditions and snow cover over large mountainous areas, suffer from this lack of ob-
servational information. The use of numerical snow models provides a more continuous spatial and temporal coverage than
observations. The complexity of such models varies widely depending on their application (Krinner et al., 2018). However, all
seasonal snow modelling systems are affected by the strong dependence of the snow cover state at any time on its past evolu-
tion since the first snowfall. This long-term dependence means that any simulation error at any time can affect all subsequent
simulations, resulting in an accumulation of errors throughout the winter.

Satellite observations of some snow properties provide a great opportunity to identify and reduce these errors (Awasthi and
Varade, 2021; Largeron et al., 2020). Methods based on ensemble data assimilation algorithms have been developed to process
these observations, (Magnusson et al., 2017; Cluzet et al., 2021). According to Cluzet et al. (2022) and Deschamps-Berger
et al. (2022), these methods primarily use snow observations to compensate for errors in the precipitation forcing of the snow
cover model. Quantifying the uncertainties in the precipitation fields is therefore essential to fully benefit from the assimilation
of snow observations. However, all the papers cited above rely only on stochastic perturbations of the precipitation dataset,
obtained with homogeneous and rather arbitrary error estimates in the absence of more advanced quantification of precipitation
uncertainty.

Existing snow cover modelling systems mostly use precipitation provided by numerical weather prediction (NWP) output,
surface observations or a combination of these two sources of information (Morin et al., 2020). Surface observations pro-
vide reliable local estimates of precipitation. However, the under-sampling of higher elevations (Thornton et al., 2022) means
that there is a lack of information on the spatial distribution of elevation-dependent variables, such as precipitation (Mott
et al., 2023). On the contrary, high-resolution NWP models produce spatialised estimates of precipitation fields at different
spatio-temporal resolutions. Lundquist et al. (2019) argue that such models can simulate annual precipitation accumulation in
mountainous areas better than estimates from gauge or radar-based observations. However, they suffer from biases and po-
sitioning errors in individual events and in seasonal accumulations. These errors are problematic for snow cover modelling
(Vionnet et al., 2016, 2019; Haddjeri et al., 2023). A combination of surface observations and NWP output is used in some
operational snow modelling systems (SAFRAN, Durand et al., 1993; Lespinas et al., 2015) to provide precipitation estimates
at scales of a few hundred square kilometres. However, scarce observations may be insufficient to constrain high-resolution
precipitation analyses in mountainous areas (Soci et al., 2016), even when specifically designed for this purpose (Schirmer and





Jamieson, 2015).

Radar measurements provide high resolution spatial estimates of precipitation. However, they are subject to uncertainties in mountainous regions, mainly due to the interaction between the radar beam and the terrain (ground clutter and partial

masks, Germann et al., 2022; Foresti et al., 2018; Faure et al., 2019; Yu et al., 2018; Foehn et al., 2018; Ghaemi et al., 2023). Methods have been developed to correct radar-based precipitation fields (Vogl et al., 2012) and to assess the associated uncertainty (Kirstetter et al., 2010; Villarini et al., 2014; Kirstetter et al., 2015). However the low quality of precipitation estimates based on radar measurements in complex terrain, even when combined with in-situ observations (Sideris et al., 2014; Sivasubramaniam et al., 2019; Champeaux et al., 2009), currently prevents their direct use to successfully force a snowpack

model (Haddjeri et al., 2023). More sophisticated products combining NWP outputs, surface observations and precipitation estimates from radar measurements (CaPA, Fortin et al., 2015, 2018; Khedhaouiria et al., 2022) suffer from significant biases in winter (Lespinas et al., 2015). The potential of using radar observations for detailed snowpack modelling has only been investigated on a relatively large scale over the French Alps (Birman et al., 2017). Over the Austrian Alps, the SNOWGRID system (Olefs et al., 2013) uses radar observations via the INCA now-casting system (Haiden et al., 2011) to force a simple

snowpack model.

As noted above, Cluzet et al. (2022) and Deschamps-Berger et al. (2022) showed that any precipitation analysis designed for a snow cover modelling system with assimilation of snow observations must include an estimate of the precipitation analysis errors. This can be done through ensemble analysis and benefit from ensemble versions of NWP models developed specifically for this purpose. A variety of methods have been developed to produce ensembles of estimated precipitation from radar and

gauge measurements with varying degrees of complexity (Clark and Slater, 2005; Germann et al., 2009; Mandapaka and Germann, 2010; Dai et al., 2014; Kirstetter et al., 2015; Frei and Isotta, 2019), but they do not use NWP outputs. Ensemble methods combining radar-based precipitation estimates and NWP output are more common in the now-casting context (Foresti et al., 2012; Nerini et al., 2019; Atencia et al., 2020a; Sideris et al., 2020).

The aim of this study is to explore the combination of different products based on radar, gauges and NWP data (section 2) to

produce ensemble precipitation analyses (section 3.2) over mountainous areas. This preliminary study focuses specifically on the French Alps, but the proposed methodology can be applied to any mountainous area with at least one year of radar-based and NWP model precipitation estimates. These analyses are expected to appropriately quantify precipitation uncertainties in order to guarantee the efficiency of assimilation of snow observations in snow cover simulations. An evaluation of the quality of several available precipitation estimation products is first performed (section 4.1). Then, a pre-processing step of the best

product is proposed to remove spatial artefacts. Finally, this study develops three different methods for ensemble analysis of daily precipitation to investigate the benefits of combining observational precipitation estimates and NWP outputs. These methods are then applied to produce ensemble analyses of daily precipitation at a 1km resolution (see section 3.2). Section 4.2 evaluates their performance and section 5.3 discusses their respective advantages and disadvantages.



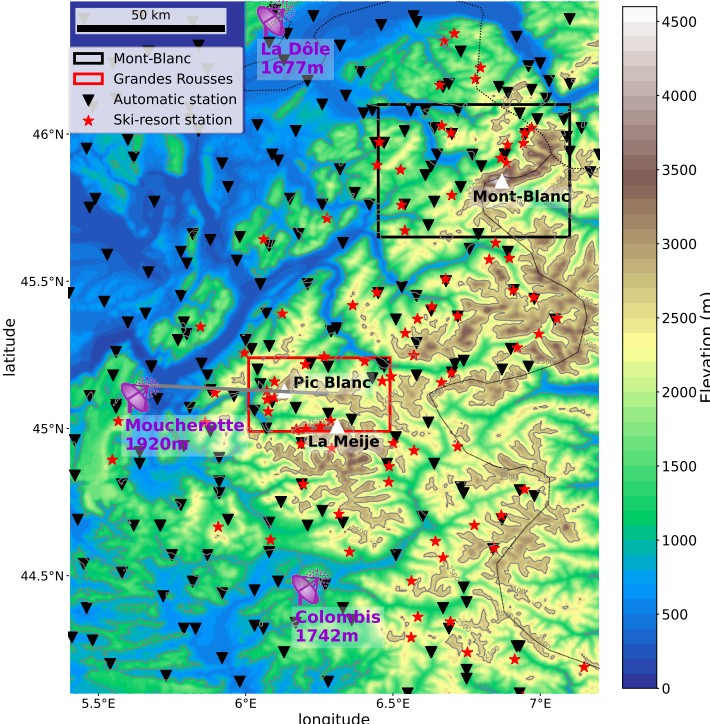

**Figure 1.** Relief at 250 m resolution of the French Alps domain used in this study, showing the three radars used in radar-based precipitation estimation products, the automatic observation stations and the reference ski-resort observation stations used for verification. The Grandes-Rousses and Mont-Blanc areas, on which parts of this study are focused, are framed and the cross section of Figure 2 is marked with a grey line.

## 2 Precipitation dataset

This study focuses on the French Alps region (Figure 1) for the period from 1 August 2021 to 1 August 2022. Evaluation data is only available for the period from 1 December 2021 to 30 April 2022.

Three different observational precipitation estimates and one from a high-resolution NWP model were considered. To avoid confusion between snow variables, we follow the international classification for seasonal snow on the ground (Fierz et al., 2009) and express all precipitation in $\mathrm{kg\,m^{-2}}$.

### 2.1 Radar product (PANTHERE)


PANTHERE (Tabary, 2007; Figueras i Ventura and Tabary, 2013) is an operational Quantitative Precipitation Estimation (QPE) product based on the combination of most of the French metropolitan radar data. It has a horizontal resolution of 1x1 $\mathrm{km^2}$ and a temporal resolution of 5 minutes, aggregated in this study into 24 h precipitation accumulations at 8:00 CET each day.



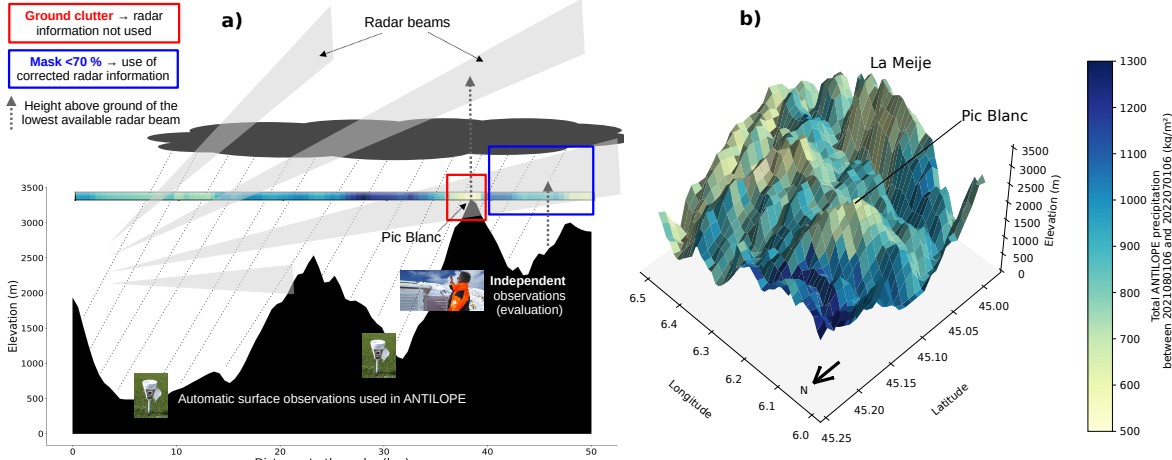

**Figure 2.** a) Illustration of radar measurement issues from the Moucherotte radar towards the Pic Blanc in the Grandes Rousses massif (grey line in Figure 1). Radar beams from four elevation angles are shown to illustrate the vertical sampling of the atmosphere and its limitations in complex terrain. The effect of ground clutter management in the PANTHERE algorithm and the link with the underestimation of precipitation over mountain ridges is illustrated over the Pic Blanc (framed in red). The position of two automatic gauges and a ski-resort observation site are also shown to illustrate the altitudes of typical in-situ observations used in the ANTILOPE product and of the reference observations used in this study, as well as the lack of in-situ observations at altitudes above 2000 m. b) ANTILOPE precipitation accumulation between 1 August 2021 and 1 July 2022 over the relief of the Grandes Rousses domain (framed in red in Figure 1).

Each radar measures the reflectivity and dual polarisation variables with a resolution of 240 m × 0.5°, up to a maximum range of 255 km, and at several elevation angles (see grey radar beams in Figure 2a). After a correction step to account for measurement problems, the reflectivity Z is converted to an instantaneous precipitation rate R ($\mathrm{kg\,m^{-2}\,h^{-1}}$) using a Z-R relationship (Marshall and Palmer, 1948) that is constant in space and time. This precipitation rate is then corrected by applying a Vertical Profile of Reflectivity (VPR) correction factor. The purpose of this step is to correct for the expected variation in reflectivity with altitude due to different hydrometeor types, and in particular to correct for the increase in reflectivity in the melt region, known as the "bright band". Although this method is effective in most cases, it has some limitations: it assumes a constant precipitation rate below the bright band, so processes such as evaporation or low-level enhancement of precipitation must be considered (Le Bastard et al., 2019).

After this processing of the volumetric radar data, the precipitation rate at ground level is estimated from a combination of colocated precipitation rates measured at all heights, weighted by their quality index, which depends on, among other factors, the height of the measurement. The final 5-minute QPE at ground level is then obtained by accumulating the precipitation rates over time. In this study, precipitation accumulations over 24 h are considered. Despite all these steps to calculate the QPE, its quality in space is variable. In general, the uncertainty of the estimate increases with the height of the radar beam above the ground, implying that the most valuable radar information comes from the lowest radar heights and that the quality of the





precipitation estimate tends to decrease far away from the radar. In particular, Figure 2a illustrates that stratiform precipitation
systems are not detected by radar beams above the top of precipitation clouds.

In mountainous areas, the radar beam will also often intercept the ground. Radar elevations affected by ground clutter are
rejected by the algorithm (red frame on Figure 2) and the lowest information comes from the next beam above, which affects
the quality of the precipitation estimate. In addition, the conical shape of the radar beam means that the beam width increases
with distance from the radar (up to about $1000\,\mathrm{m}$ at a distance of $50\,\mathrm{km}$). It can also be partially affected by the presence of a
mountain (blue frame in Figure 2). In this case, the affected beam information is considered unusable behind the mountain if
the mask blocks more than 70% of the total energy, otherwise the signal is corrected for attenuation.

Faure et al. (2017) evaluated the quality of PANTHERE precipitation estimation over the French Alps, and showed an
increasing underestimation of precipitation towards the east due to radar beam blockage and increasing distance from radars.
They also identified specific areas affected by significant underestimations related to ground clutter handling (as shown in
Figure 2) and concluded that the clutter correction is ineffective in a high mountain context. Similarly, Faure et al. (2019)
studied the vertical distribution of PANTHERE precipitation estimates and highlighted a general overestimation of the radar
QPE at the bottom of the valleys and an underestimation at the highest altitudes.

## 2.2    Radar and gauges fusion product: ANTILOPE

ANTILOPE (Champeaux et al., 2009) is an operational composite analysis combining radar precipitation estimates from
PANTHERE and precipitation observations from automatic gauges (see Figure 1). It is available at a resolution of $1\mathrm{x}1\ \mathrm{km}^2$
and $24\,\mathrm{h}$ accumulations at 8:00 CET have been used in this study. The fusion of these two sources of information is based on
a scale separation between small scale convective and large scale stratiform precipitation. Precipitation associated with radar-
detected convective cells is corrected by a spatialisation of the local differences between radar and gauge precipitation estimates
using inverse distance interpolation. Large scale precipitation is estimated by a spatialisation of the gauge values by an ordinary
kriging with external drift method, using either a correlogram computed from radar images or an exponential variogram model
if no radar image is available. In contrast to PANTHERE, the quality of ANTILOPE precipitation estimation in mountainous
areas is not well documented. The strong dependence of the ANTILOPE product on the radar-based precipitation estimate
suggests that the main drawbacks of radar measurements described in section 2.1 also affect the ANTILOPE quality. However,
the use of rain-gauge observations may reduce the errors' magnitude, with possible exceptions in case of assimilation of non-
heated gauges not detected by the control steps.

## 2.3    Gauge kriging

In order to document the added value of the radar information used in ANTILOPE, a precipitation estimation based on the same
kriging method (using an exponential variogram) of the same set of gauges (automatic stations, Figure 1) as those used in the
ANTILOPE algorithm, but without any radar information was set up and evaluated for this study. The resulting precipitation
fields have the same $1\,\mathrm{km}$ resolution as the ANTILOPE and PANTHERE products and are used similarly. The temporal
resolution used in this study is $24\,\mathrm{h}$.



## 2.4 Evaluation data

The evaluation dataset comes from the ski-resort observation network of the ski resorts of the French Alps (Fig. 1). These observations are not used by the ANTILOPE product (and therefore not used in gauge kriging). This network provides a set of daily man-made meteorological and snowpack observations specifically designed for avalanche forecasting during the winter season (generally from mid-December to mid-April, depending on the opening and closing dates of the resorts). For this study, observations of 24 h precipitation accumulation in a bucket weighted at 8:00 (CET) are used as the reference for all evaluations. Consequently, all precipitation products are evaluated in terms of 24 h water equivalent accumulations starting at 8:00 CET. A human estimate of the highest altitude reached by the rain-snow limit during the same period is also available and used in this work. When focusing on solid precipitation, only days where the rain-snow limit altitude was below the station altitude were considered.

Ski-resort observations have limitations for the evaluation of gridded precipitation estimates :

- Human measurement time may vary slightly between stations and days

- the mountainous environment is known to affect the measurements from the gauges, with possible snow accumulation in the gauges or under-catchment in windy conditions (these measurement errors are sometimes detected and corrected or removed in the monitoring process of the observations and also affect the automatic gauges used by the ANTILOPE product)

- local gauge observations do not have the same representativeness as a gridded estimate over a pixel of about $1{\times}1$ km$^2$

## 2.5 Numerical weather prediction model

In this study, data from the deterministic NWP model AROME (Seity et al., 2011; Brousseau et al., 2016) and its ensemble version have been used in a complementary way.

The French operational high-resolution NWP model AROME provides hourly precipitation forecasts with a resolution of $1{\times}1$ km$^2$. Daily precipitation accumulations (liquid and solid) are directly derived from the 24 h forecast at validity time 6:00 UTC. Yearly AROME precipitation accumulation is then calculated as the accumulation of daily precipitation between 1 August 2021 and 1 August 2022.

A 16-member ensemble version of the AROME NWP model is also operational at a resolution of $2.5{\times}2.5$ km$^2$ (Bouttier et al., 2016). Its precipitation forecasts are statistically post-processed with the method developed by Taillardat et al. (2019), based on quantile regression forests, to provide an unbiased and well-distributed ensemble of hourly precipitation forecasts, hereafter referred to as PEAROME. The calibration uses ANTILOPE precipitation estimates as reference and is performed at two AROME EPS initialisation times (9:00 UTC and 21:00 UTC) with lead times up to 45 h. (Taillardat and Mestre, 2020). Physically realistic precipitation patterns are then reconstructed using the ensemble copula coupling method of Schefzik et al. (2013). However, the training method has been evaluated over metropolitan France (Taillardat and Mestre, 2020) and its performance over the French Alps is not well known. The daily precipitation used in this study is the precipitation accumulation





between lead times 9 h and 33 h from the 21:00 UTC initialisation time. The raw precipitation fields are downscaled to the
ANTILOPE $1 \times 1$ km$^2$ grid by a bi-linear interpolation so that model and observation can be compared at the same spatial and
temporal resolution.

## 3 Method

### 3.1 ANTILOPE observation error

A precipitation analysis requires the specification of the error of the various products involved. As mentioned earlier, radar-
based observations are known to suffer from important shortcoming in mountainous areas (Germann et al., 2022). In particular,
unrealistic spatial patterns of ANTILOPE yearly precipitation accumulation estimates can be visually identified over some
mountain ridges. Figure 2b shows that ANTILOPE yearly precipitation accumulations over the ridges of Pic-Blanc and La
Meije (around 600 kg m$^{-2}$) are only about half those over the lower elevations areas in between (up to 1300 kg m$^{-2}$). Figure
3a shows the same pattern over the Mont-Blanc (circle in red) where yearly precipitation accumulation (less than 200 kg m$^{-2}$
per year, five times less than in the Chamonix Valley circle in green) is unrealistically low. These patterns are also visible in
the daily precipitation fields (Figure 5) and are probably due to the presence of ground clutter (see section 2.1).

The most common approach to overcome the limitations of radar-based precipitation estimates in mountainous areas is
to combine them with other sources of information or to apply a calibration step (Germann et al., 2022). However, these
methods suffer from the lack of observations at high elevations (Figure 1). McRoberts and Nielsen-Gammon (2017) proposed
a method to detect and correct pixels affected by partial beam blockage based on an analysis of the radar precipitation estimation
climatology, but it does not specifically focus on ground clutter, which is prevalent in mountainous areas. Haiden et al. (2011)
uses radar precipitation estimates where ground clutters have been statistically filtered, and other perturbations are dealt with in
a preprocessing step bases on a climatological scaling of the radar data. Here, we propose a method to mitigate the unrealistic
spatial patterns in ANTILOPE precipitation fields. A first method to estimate the climatological errors of ANTILOPE is
described in section 3.1.1 and an evaluation of the method is provided in section 4.2. This estimated climatological error is
then used to remove the unrealistic spatial structures in ANTILOPE 24 h precipitation fields (Section 3.1.4).

### 3.1.1 ANTILOPE climatological error estimation

We propose a method based on a comparison between ANTILOPE, the automatic gauge measurements (whether used in
ANTILOPE or not) and the AROME yearly precipitation accumulations to provide :

– an estimate $\hat{R}$ of the real climatological ANTILOPE ratio ($R^{\text{TRUE}}$) defined as the ratio between the ANTILOPE pre-
cipitation accumulation (RR$^{\text{ANT}}$) and the real precipitation accumulation (RR$^{\text{TRUE}}$, unknown) over the same period :

$$R^{\text{TRUE}} = \frac{\text{RR}^{\text{ANT}}}{\text{RR}^{\text{TRUE}}} \qquad (1)$$



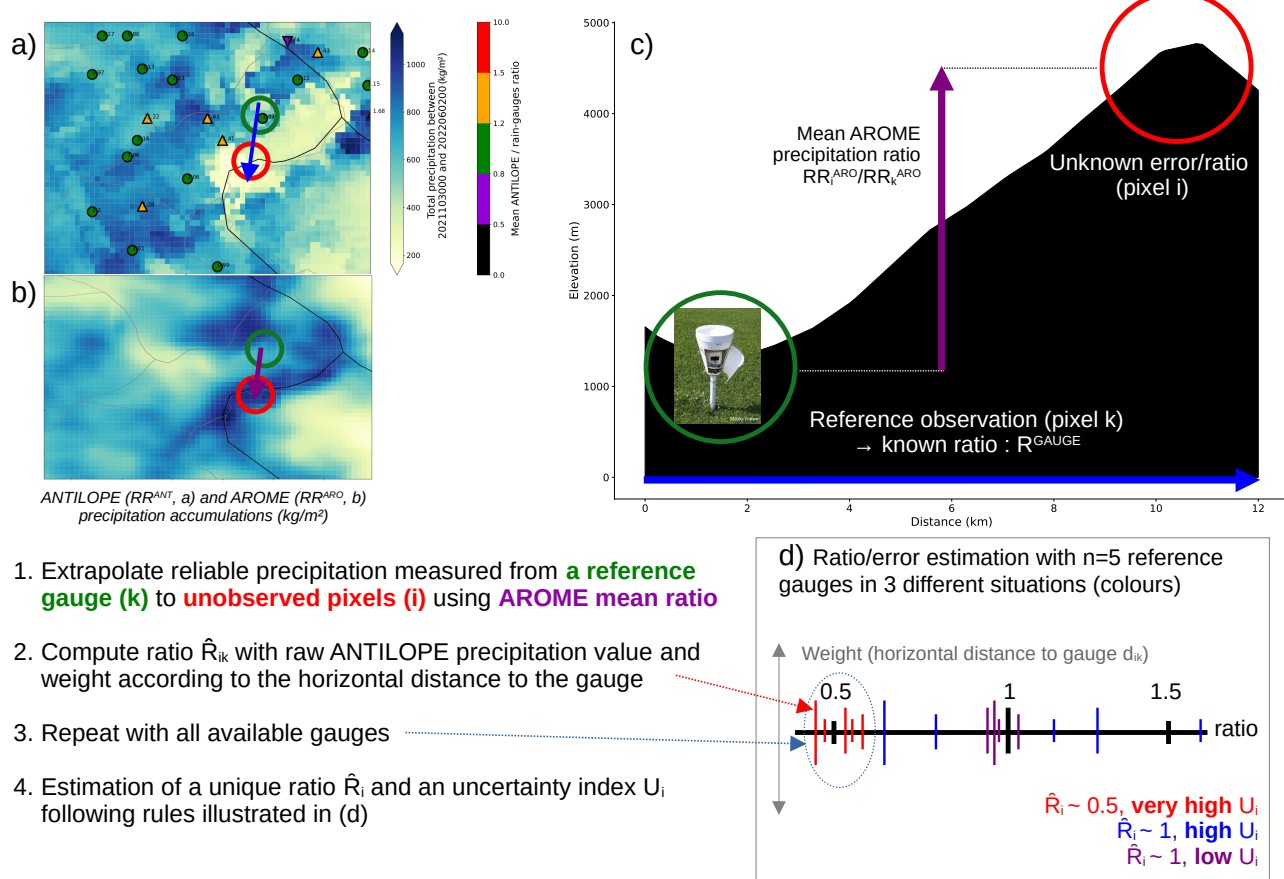

**Figure 3.** Illustration of the climatological ANTILOPE error estimation over the Mont-Blanc (circled in red) based on (a) the Yearly ANTILOPE precipitation accumulation ($RR^{ANT}$) from 30 October 2021 to 2 June 2022 and the mean ANTILOPE / gauge ratio ($R^{GAUGE}$) observed at automatic stations and (b) the AROME precipitation accumulation ratio between the gauge pixel and the Mont-Blanc pixel ($RR_i^{ARO}/RR_k^{ARO}$, purple arrow).



Where a gauge is available, we assume $RR^{GAUGE} = RR^{TRUE}$ so that:

$$\hat{R} = R^{GAUGE} = \frac{RR^{ANT}}{RR^{GAUGE}} \tag{2}$$

If $\hat{R}$ can be spatialized, it can be used as a correction factor for daily precipitation fields in order to remove systematic errors

– a spatialized ANTILOPE uncertainty index, which measures the relative confidence between the different pixels of the domain

The general idea of the method is to extrapolate the ratios measured at gauges locations to any location without reference observation considering precipitation accumulations simulated by the AROME NWP model, which are considered to better represent mean vertical gradients. The estimation method is illustrated on Figure 3 over the Mont-Blanc pixel (hereafter referred to with the subscript $i$ and circled in red on Figure 3a,b,c).

### 3.1.2 Ratio estimation

The estimation of the ratio $\hat{R}_i$ for any pixel $i$ (e.g. Mont-Blanc point in Figure 3) is based on :

– the ratio between the ANTILOPE yearly precipitation accumulation at pixel $i$ $RR_i^{ANT}$ and the ANTILOPE accumulation $RR_k^{ANT}$ at nearby gauges $k$ (circled in green in Figure 3a,b,c), far from pixel $i$ of a distance $d_{ik}$;

– the ratio between the AROME yearly precipitation accumulations $RR_i^{ARO}$ and $RR_k^{ARO}$ over pixels $i$ and $k$ (indicated by the purple arrow on Figure 3b,c)

$$\hat{R_{ik}} = \frac{RR_i^{ANT}}{RR_k^{ANT}} \cdot \frac{RR_k^{ARO}}{RR_i^{ARO}} \tag{3}$$

Considering all $k \in [1, n]$ nearby gauges (Figure 3a) and giving them a weight proportional to the distance $d_{ik}$ from pixel $i$ (e.g. Mont-Blanc) :

$$w_{ik} = 1 - \frac{d_{ik}}{d_0} \tag{4}$$

(where $d_0 = 20\,\mathrm{km}$ is the distance range parameter, arbitrarily chosen based on the density of the surface observation network), a weighted ensemble of estimated ratios is obtained (bars of the same colour in Figure 3d).

The weighted mean ($M_i$) and the spread ($S$) of this weighted ensemble of estimated ratios for pixel $i$ are :

$$M_i = \frac{1}{W_i} \sum_{k=1}^{n} w_{ik} \hat{R_{ik}} \tag{5}$$

and

$$S_i = \sqrt{\frac{1}{W_i} \sum_{k=1}^{n} w_{ik}(\hat{R_{ik}} - M_i)^2} \quad \text{where} \quad W_i = \sum_{k=1}^{n} w_{ik} \tag{6}$$

The conversion of this weighted ensemble into a single ratio value $R_i$ is illustrated in Figure 3d and follows the idea that :





- The lower the weighted spread $S_i$ (i.e. the ratios $\hat{R_{ik}}$ estimated with the $n$ nearby gauges are similar) the closer the estimated ratio $\hat{R_i}$ is to the weighted ensemble mean $M_i$.

- The higher the weighted spread $S_i$ (i.e. the ratios $\hat{R_{ik}}$ estimated with the $n$ nearby gauges give divergent information), the closer the estimated ratio $\hat{R_i}$ is to 1 (no relevant information can be derived, so no correction is applied). This situation is illustrated by the blue bars in Figure 3d.

Thus, the estimated ratio over the pixel $i$ (the Mont-Blanc in the example) is given by :

$$\hat{R_i} = K_i \times M_i + 1 \times (1 - K_i) \quad \text{where} \quad K_i = e^{-\frac{S_i \times \ln(10)}{|M_i - 1|}} \tag{7}$$

$K_i$ ensures that the estimated ratio lies between 1 and the weighted mean ($M_i$) depending on the spread $S_i$ of the estimated ratio values. The constant exponential decay is fixed to ensure that if the weighted spread of the ensemble of estimated ratios among the $n$ gauges ($S_i$) is high, the confidence that $M_i$ is a good estimator of $R_i^{\text{TRUE}}$ is low.

### 3.1.3 Uncertainty index estimation

The associated climatological uncertainty index $U_i$ is defined so that it increases with:

- $|\hat{R_i} - 1|$, with a magnitude depending on the uncertainty associated with the estimation method, measured by $K_i$.

- the discrepancies between the individual ratio estimates associated with each gauge, which increase with $S_i$ and $|M_i - 1|$.

The relationship between ANTILOPE errors and $R^{\text{TRUE}}$ was estimated over rain gauge stations by computing a linear regression between ANTILOPE error compared to reference gauge measurements and the ratio $R^{\text{GAUGE}}$ between ANTILOPE and reference gauges (see figure A1). This statistical relationship is used with the other estimators $\hat{R_i}$ and $M_i$ to estimate an uncertainty $U_i$ at each point of the domain.

$U_i$ is thus defined as a sum of two terms :

$$U_i = (1 + K_i) \times A_i + (1 + S_i) \times B_i \tag{8}$$

where

$$A_i = \begin{cases} -20.137 \times (\hat{R_i} - 1) & \text{if } \hat{R_i} < 1 \\ 16.787 \times (\hat{R_i} - 1) & \text{if } \hat{R_i} \geq 1 \end{cases} \quad \text{and} \quad B_i = \begin{cases} -20.137 \times (M_i - 1) & \text{if } M_i < 1 \\ 16.787 \times (M_i - 1) & \text{if } M_i \geq 1 \end{cases} \tag{9}$$

In extreme cases:

- when all estimated $\hat{R_{ik}}$ are identical, $S_i = 0$ and $K_i = 1$, so $U_i$ is obtained from $A_i$ and proportional to $|\hat{R_i} - 1|$.

- on the contrary, when the estimated $\hat{R_{ik}}$ are very contrasted (large $S_i$ and $K_i$ close to 0), $U_i$ is mainly obtained from $B_i$ and proportional to $|M_i - 1|$.

Figure 4 shows the ratio (Figure 4a) and uncertainty (Figure 4b) fields estimated with this method using data from winter 2021/2022. The yearly accumulation field after correction with the estimated ratio is shown in Figure 7b.



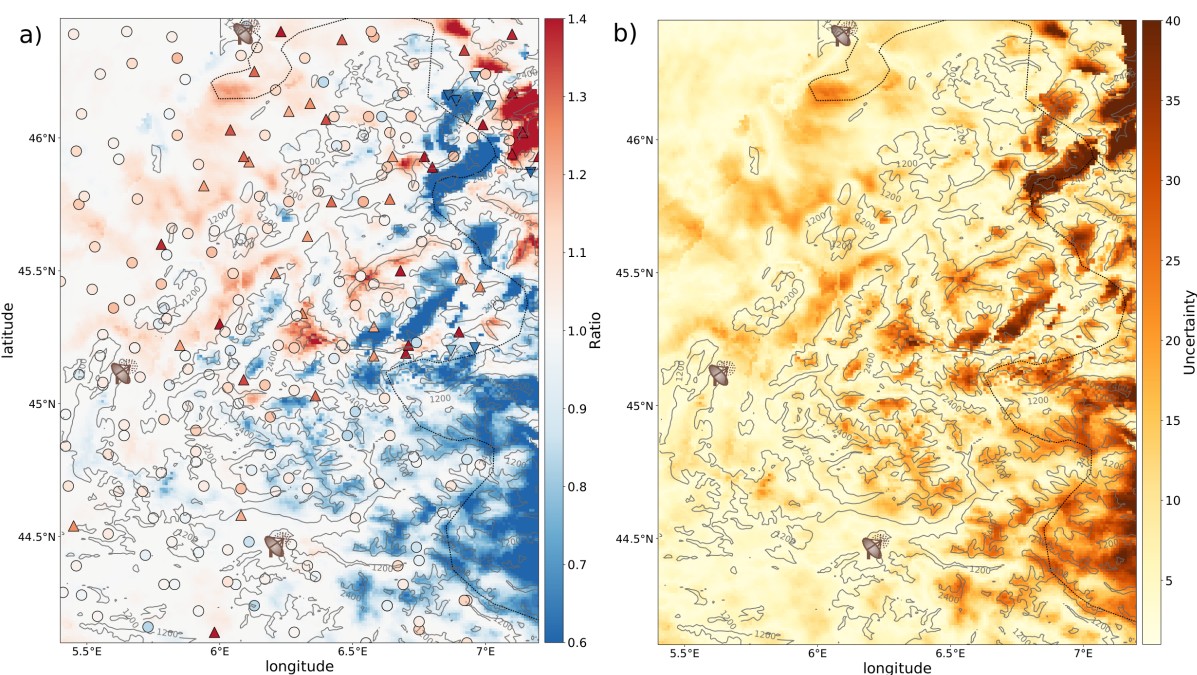

**Figure 4.** (a) Estimated climatological ratio ($\hat{R}$) based on the mean ratios between ANTILOPE and automatic gauge precipitation measurements and (b) uncertainty index ($U$) obtained using the method described in section 3.1.1. The grey lines indicate the 1200 m, 2400 m and 3600 m isolevels to illustrate the relationship between the estimated ratio/error and the relief.

### 3.1.4 Dynamic correction of ANTILOPE daily precipitation fields

To deal with the unrealistic spatial patterns described in Section 3.1, a preprocessing step is applied to the daily ANTILOPE precipitation estimation fields and is illustrated in Figure 5. A first correction is applied to mitigate climatological errors using the estimated climatological ratio field obtained in Section 3.1.1 (Figures 4a and 5b). For each pixel $i$ of the domain with a precipitation value $\mathrm{RR}_i^{\mathrm{ANT}}$ estimated by ANTILOPE (Figure 5a), the precipitation value after climatological correction (Figure 5c) is given by $\mathrm{RR}_i^d = \frac{\mathrm{RR}_i^{\mathrm{ANT}}}{R_i}$.

A modified Weighted Moving Average (WMA) filter is then applied using the uncertainty index (Figures 4b and 5c) obtained with the method described in Section 3.1.1. The WMA window includes all pixels within a $d_0 =$20 km radius from pixel $i$ (red circle in Figure 5c). To estimate the filtered precipitation of pixel $i$, each pixel $k$ within this window is assigned a weight $c_{ik} = \frac{1 - d_{ik}/d_0}{U_k/U_0}$, where $d_{ik}$ is the distance to the pixel at the centre of the moving window, $U_k$ is the uncertainty index of pixel $k$ (see Section 3.1) and $U_0 = 1 \ \mathrm{kg\,m^{-2}}$ a normalisation factor. Considering all $n$ pixels of the window, the weighted mean





precipitation ($\overline{\mathrm{RR}_i^d}$) and the variance ($V_i$) of all weighted precipitation values over the window associated to pixel $i$ are :

$$\sim \begin{cases} \overline{\mathrm{RR}_i^d} = \frac{1}{C_i} \sum\limits_{k=1}^{n} c_{ik} P_k^d \\ V_i = \frac{1}{C_i} \sum\limits_{k=1}^{n} c_{ik} (\mathrm{RR}_{ik}^d - \overline{\mathrm{RR}_i^d})^2 \end{cases} \quad \text{where} \quad C_i = \sum_{k=1}^{n} c_{ik} \tag{10}$$

While a standard WMA filter would have considered $\overline{\mathrm{RR}_i^d}$ as the new precipitation estimate for pixel $i$, in this work the filtered value is combined with the local value with respective weights depending on the estimated spatial error (Figure 5e):

$$\mathrm{RR}_i = \frac{\mathrm{RR}_i^d c_{ii} + \overline{\mathrm{RR}_i^d} \frac{C_i}{n}}{c_{ii} + \frac{C_i}{n}} \tag{11}$$

This weighted correction ensures that a pixel with a low estimated spatial error ($c_{ii} >> \frac{C_i}{n}$) is not affected by the WMA filter while a pixel with a high estimated spatial error ($c_{ii} << \frac{C_i}{n}$) is closer to the average precipitation value in the localisation area.

The observation error (Figure 5f) is finally defined as the weighted spatial standard deviation of the daily precipitation over this window :

$$E_i = \sqrt{V_i} \tag{12}$$

## 3.2 Ensemble analysis

Three different methods are proposed to produce ensemble precipitation analyses of daily precipitation fields based on the pre-processed ANTILOPE precipitation fields obtained with the algorithm described in Section 3.1.4. Two of these methods are based on ensemble data assimilation algorithms that combine an observation (pre-processed ANTILOPE precipitation field) and an ensemble of NWP model outputs (post-processed PE-AROME precipitation fields).

### 3.2.1 Random Sampling

The first ensemble analysis method uses output fields of the ANTILOPE preprocessing step described in Section 3.1.4. 16 precipitation fields are constructed by random perturbations around the corrected precipitation field $\widetilde{P^i}$ (Figure 5e).

We have chosen to separate the spatial and dynamical components of the observation error in a similar way to Villarini et al. (2014). The magnitude of the perturbation is thus determined by two terms :

– a spatial observation error $E_i$ resulting from the preprocessing step (Section 3.1.4), which accounts for the spatial variability of the ANTILOPE quality described in Section 3.1.

– a dynamic error expressed as a fraction of the precipitation intensity. The magnitude of this error has been estimated from the linear regression between the ANTILOPE precipitation intensity and the associated error (not shown) and is set at 30% of the precipitation intensity.

The random perturbation for a given ensemble member $m$ is then determined by two different values :



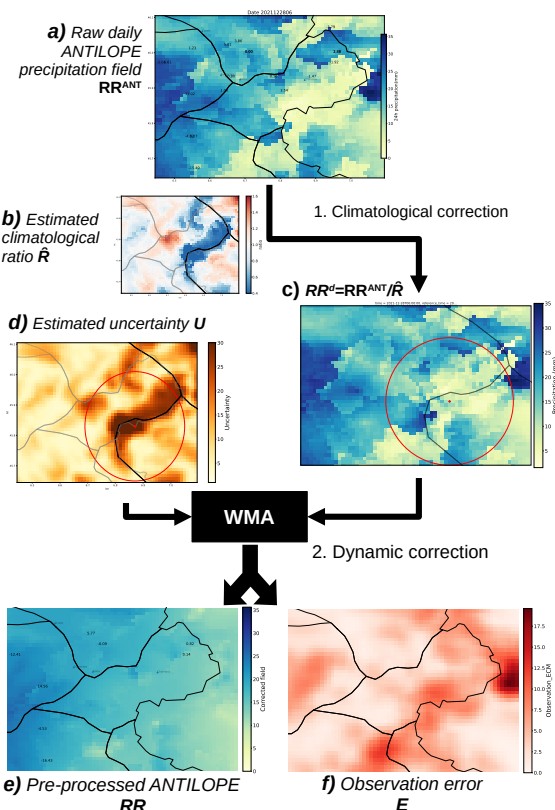

**Figure 5.** Illustration of the ANTILOPE pre-processing step applied for daily precipitation fields. First the climatological ratio ($\hat{R}$) estimated with the method described in Section 3.1.1 is used as a climatological correction factor to mitigate systematic errors. The resulting precipitation field ($RR^d$) is then filtered with a modified Weighted Moving Average (WMA) method, which produces a smoother output daily precipitation field (RR) as well as an associated error field ($E$). The weights used in the WMA step are a combination of an inverse distance weighting and the uncertainty index ($U$) estimated using the method described in Section 3.1.1.





- $N^m$ is sampled from a normal distribution centered on 0 and with a variance of 1 and applied as a multiplicative factor to the spatial component of the error

- $G^m$ is sampled from a gamma distribution (with a shape parameter $k = 2$ and a scale parameter $\theta = \sqrt{1/k}$ ensuring a variance of 1) shifted by $k\theta$ so that the mean of the distribution is 0

For each pixel $i$ of the domain with a corrected precipitation $\mathrm{RR}_i$ (equation 11) and an associated error $E_i$ (Equation 12), the precipitation in member $m$ of the output ensemble is thus :

$$\mathrm{RR}_i^m = \max(\mathrm{RR}_i(1 + 0.3G^m) + N^m.E_i, 0) \tag{13}$$

Drawing random values from distributions with a mean centred on 0 ensures that the resulting ensemble mean is expected to be $\mathrm{RR}_i$, unless the magnitude of the spatial error is greater than the precipitation value itself (in which case the $0 \ \mathrm{kg\,m^{-2}}$

lower bound may shift the ensemble mean upwards).

This Random Sampling (RS) analysis is the direct translation of the ANTILOPE pre-processing step, which produces a corrected precipitation field and associated error, into an ensemble analysis. It can be considered as a benchmark product for ensemble analysis.

### 3.2.2 Particle Filter

The second ensemble analysis is based on particle filter theory. Particle Filters (PF) are sequential Monte Carlo algorithms commonly used for data assimilation in non-linear systems (van Leeuwen, 2009). These methods are based on the approximation of the model probability density function (PDF) by an ensemble of model states (called ensemble members and denoted $X$ in the following equations). When an observation $\mathrm{RR}_i$ is available, this ensemble of model states is updated in two steps. First, the different members are weighted according to their likelihood (distance to the observation). Then, the model state PDF

is updated by resampling the different members according to their weights: members with high weights are replicated, while those with lower weights are dropped. To deal with the known problems of the PF algorithm for large numbers of observations, we chose to apply the particle filter algorithm for each pixel independently, reducing the problem dimension to 1 observation for a 16-member ensemble, as suggested by Snyder et al. (2008). Thus, the likelihood of the PE-AROME precipitation forecast $X_i^m$ of member $m$ over a pixel $i$ can be simply formulated as a function of the corresponding ANTILOPE pre-processed

precipitation $\mathrm{RR}_i$ (Equation 11) and its associated error $E_i$:

$$L_i = e^{-\left(\frac{(X_i^m - \mathrm{RR}_i)}{E_i}\right)^2} \tag{14}$$

The main disadvantage of this local approach is the loss of spatial consistency. In fact, the analysed precipitation accumulation of neighbouring pixels can be obtained from combinations of different members. To obtain consistent precipitation fields, a field reconstruction step based on an ensemble copula coupling method (Schefzik et al., 2013) is applied during the

re-sampling step. This re-sampling procedure is based on preserving the rank structure of the ensemble members from the



original ensemble to the analysed one, ensuring, for example, that the same member has the highest precipitation value on a pixel before and after the analysis. In this way, large-scale structures are preserved in the analysis, although local discrepancies may persist.

### 3.2.3 Ensemble Kalman Filter

The last ensemble analysis is based on the Ensemble Kalman Filter algorithm (EnKF, Evensen, 2003). EnKF are Monte Carlo implementations of the Kalman Filter (Kalman, 1960), where an ensemble of simulations is used to sample the model state (background) distribution. The main advantage over the PF is that it preserves the spatial consistency of the analysed fields. We have chosen an approach inspired by that of Atencia et al. (2020b), using only the analysis step of the EnKF, which consists in updating each ensemble member with the information coming from an observation. It is assumed that the cross-correlations

between the background and the observation are negligible, so that each pixel of the domain is processed independently. For a given pixel $i$, the precipitation estimated by the ensemble member $m$ ($X_i^m$) is pulled towards the corresponding observation (ANTILOPE pre-processed precipitation $\mathrm{RR}_i$ from equation 11):

$$\widetilde{X_i^m} = X_i^m + K_i(\mathrm{RR}_i - X_i^m) \tag{15}$$

where the so called gain factor $K_i$ defines the relative influence of the original background value and the observation, based on

the observation error $E_i$ defined in equation 12 and the background error represented by the 16-member ($N = 16$) ensemble spread $B_i = \frac{1}{N-1} \sum_{m=1}^{N} (X_i^m - \overline{X_i^m})(X_i^m - \overline{X_i^m})$ :

$$K_i = B_i(B_i + E_i)^{-1} \tag{16}$$

### 3.3 Evaluation Method

The aim of this work is to produce an ensemble precipitation analysis with a resolution of 1 km over the French Alps, which

should ideally fulfil the standard requirements expected from ensemble modelling systems (reliability and resolution). This ensemble analysis should also meet the following specific conditions:

– reduced systematic bias of the ensemble mean as compared to other existing products

– improved predictive added value of the analysis as compared to other existing products

– ensemble members feature realistic spatial structures

While the first three requirements can be evaluated with the available local independent observations of the ski-resorts network, the lack of a spatialised reference observation of the precipitation fields implies that an objective evaluation of the spatial structure of the analysed fields is not directly possible at this stage.

The different products were evaluated during the 2021/2022 winter season. Each available ski-resort observation of 24 h precipitation at 7:00 CET was compared with the corresponding 24 h precipitation estimate at 6:00 UTC on the correspond-

ing 1 km pixel for each product. First, a comprehensive evaluation of existing precipitation estimation products is provided.





For deterministic products, this evaluation focuses on systematic errors and biases as well as spatial structure issues of the precipitation field, while for the PEAROME product ensemble characteristics are also considered. As all proposed ensemble analysis methods rely on a pre-processing of the ANTILOPE precipitation estimation product, an objective evaluation of the pre-processing step and its performance compared to existing products is provided. A subjective comparison of the spatial
structures of the precipitation fields is also discussed. Finally, the three ensemble analysis methods are evaluated, and their performances are compared in terms of mean bias, spread skill and overall predictive value added.

Biases, ratios and root mean square deviations are computed for each reference station by comparing the estimated and observed 24 h precipitation values time series. For ensemble products, the estimated value is taken as the ensemble mean. The predictive added value of the different products is assessed using two probabilistic scores commonly used for ensemble
prediction systems : the Brier Score (BS, Brier, 1950) and the Continuous Rank Probability Score (CRPS, Matheson and Winkler, 1976; Candille and Talagrand, 2005). More details on these scores can be found in Appendix A. Another important aspect of ensemble forecasting is the reliability of the ensemble spread : an ensemble forecasting system is reliable if the magnitude of its spread for different events and locations matches the associated spread of the ensemble mean error. This can be inferred from a scatter plot of the ensemble spread against the ensemble mean error (Hopson, 2014).

## 4    Results

### 4.1    Evaluation of existing precipitation products

The main results of the evaluation of the existing products (PANTHERE, gauge-kriging, ANTILOPE and post-processed PEAROME) are summarised in Figure 6 showing the distributions of the root mean square deviation from the ski-resorts reference observations for all data (Figure 6a) and for observed precipitation events above 10 $\mathrm{g\,m^{-2}}$ in 24 h only (Figure 6b).
This Figure shows that the precipitation estimation from the network of automatic observation stations (KRIGING) performs better than the estimation from radar measurements only (PANTHERE) in mountainous areas. The RMSE of the combination of these two sources of information (ANTILOPE) is in the same order of magnitude as that of the KRIGING product, but is sometimes improved for precipitation events above 10 kg/m2/24h (b). Similar results are obtained when only snowfall events are considered (Figure B1). Other results (frequency of relative errors less than 20%, not shown) also support the choice of
ANTILOPE as the best precipitation product available for an ensemble analysis system, especially since ANTILOPE performs equally well when only solid precipitation is considered when only solid precipitation is considered (see Figure B1 in Appendix A). However, the spatial distribution of ANTILOPE's performance shows that even if there is no widespread systematic bias, there is a high spatial variability in its performance. Figure 7a shows that the mean ratio between ANTILOPE estimated precipitation and the corresponding ski-resort reference observations can reach both very high (above 1.5, in red) and very low
(below 0.5, in black) values, even for stations separated by only a few kilometres. Moreover, the lack of evaluation stations at high altitudes or near mountain ridges does not allow a good documentation of the spatial artefacts already discussed in section3.1.





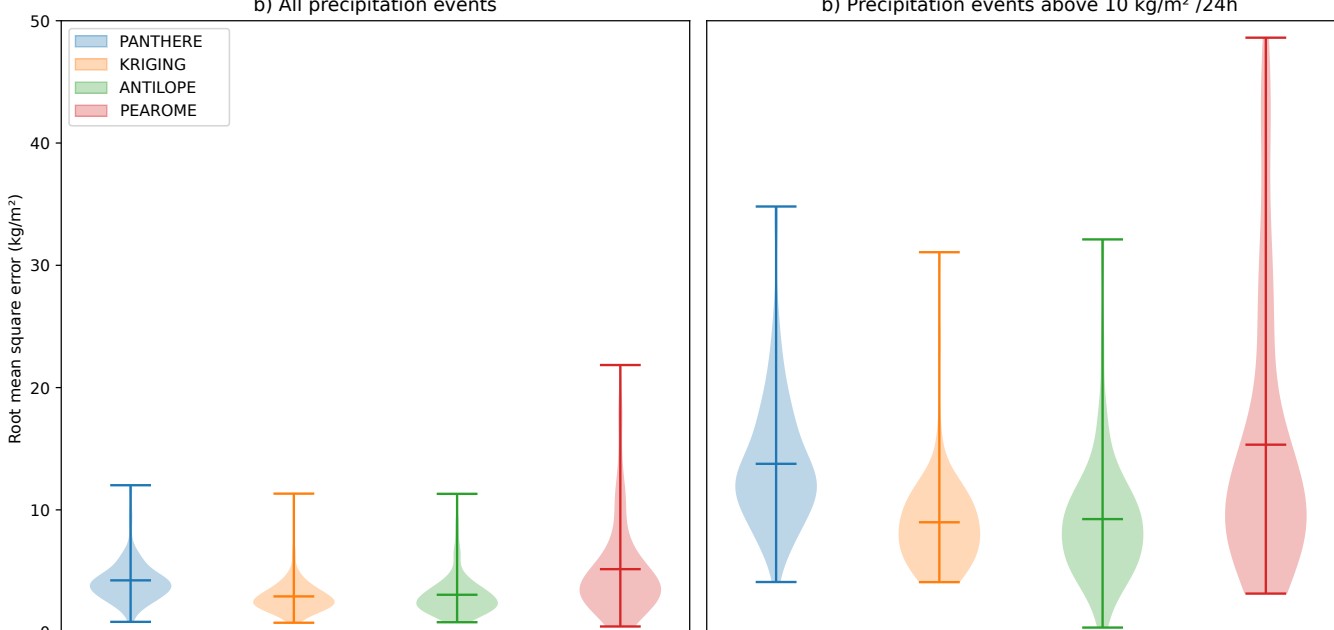

**Figure 6.** Distribution of the root mean square deviation of the existing precipitation estimation products from all ski-resort reference observations of daily precipitation (a) and daily precipitation above $10\,\mathrm{kg\,m^{-2}}$ (b) over all evaluation stations. Precipitation estimates come from radar-only observations (PANTHERE), a kriging of gauge observations (KRIGING), a fusion of radar and gauge observations (ANTILOPE), and the ensemble mean of the post-processed high-resolution ensemble NWP model (PEAROME).

Finally, the RMSE of the PEAROME ensemble mean is much higher than that of all observation-based products (see Figure

effig:rmse). This indicates that even after statistical post-processing, PEAROME forecasts provide a less reliable estimate of precipitation than observation-based products. An evaluation using metrics specifically adapted to ensemble forecasting, such as the Brier Score (BS) or the Continuous Ranked Probability Score (CRPS), shows that the probabilistic nature of PEAROME does not compensate for deficiencies in deterministic estimation.

## 4.2 Evaluation of precipitation analyses

### 400 4.2.1 Impact of the pre-processing procedure

The ensemble precipitation analyses in this study are based on the ANTILOPE preprocessing step, as explained in Section 5. This step relies on the error estimation method described in Section 3.1.1. Figure 7b demonstrates the ability of this method to identify ANTILOPE biases, showing the same information as Figure 7a but with unbiased precipitation estimates using the ratio from Figure 4a. Spatial artefacts that are visible in the yearly precipitation accumulation of Figure 7a appear to be reduced



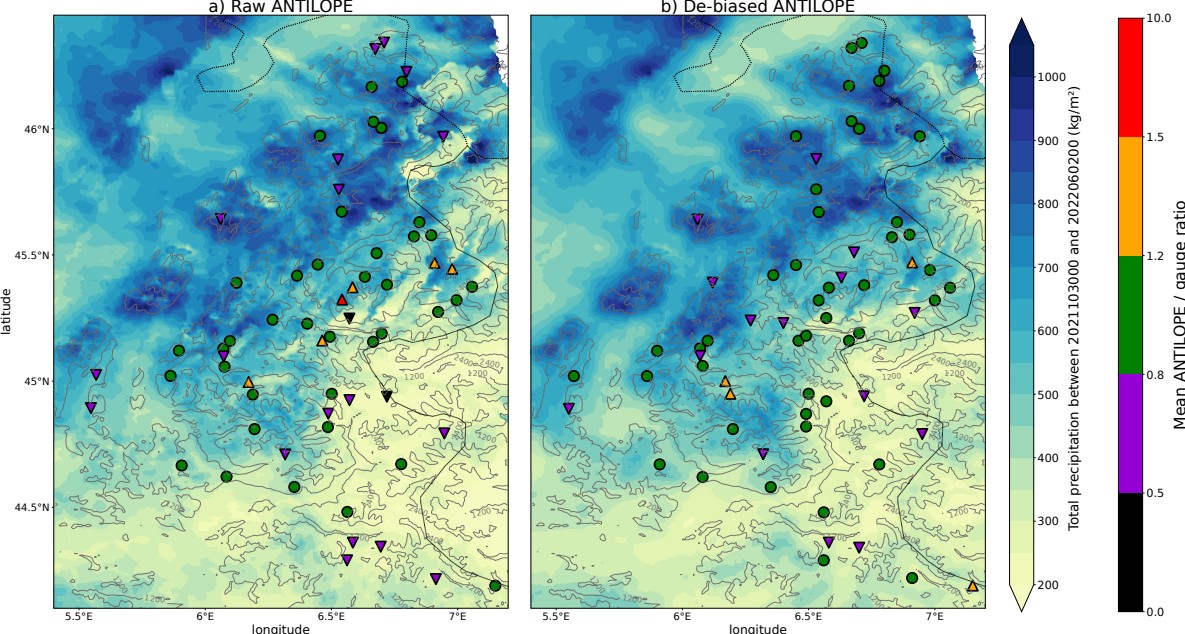

**Figure 7.** Yearly ANTILOPE precipitation accumulation over the entire French Alps domain and mean ratio with the available evaluation data from the ski-resorts network before (a) and after (b) application of the climatological correction factor obtained by the observation error estimation method.

in Figure 7b. In addition, the mean ratio with reference ski-resorts observations after climatological correction is often closer to 1 (green dots) than for raw ANTILOPE precipitation estimates.

Figures 8 and 9 show the performance of the full pre-processing step. The error estimation method reduces biases with ratios closer to 1 after the ANTILOPE pre-processing step (light blue) than before (brown), as confirmed by Figure 8. In addition, Figure 9 shows that this pre-processing step improves both CRPS and BS, regardless of the precipitation threshold.

### 4.2.2 Skill of ensemble analyses

All ensemble analyses (RS, PF and EnKF) evaluated in Figures 8 and 9 show better performance than the pre-processed ANTILOPE and PE-AROME products respectively. Figure 9a shows a clear improvement of the BS for the three ensemble analysis methods, which all have similar performance for low to moderate precipitation events (up to 10 $\mathrm{kg\,m^{-2}}$ in 24 h). However, for precipitation above 10 $\mathrm{kg\,m^{-2}}$, the basic RS method outperforms the two methods based on ensemble data analysis. For precipitation events above 25 $\mathrm{kg\,m^{-2}}$, the PF and EnKF analyses do not perform significantly better than the deterministic ANTILOPE pre-processing. This result is confirmed by the CRPS values (Figure 9), which are significantly lower for the RS analysis than for the PF and EnKF analyses.





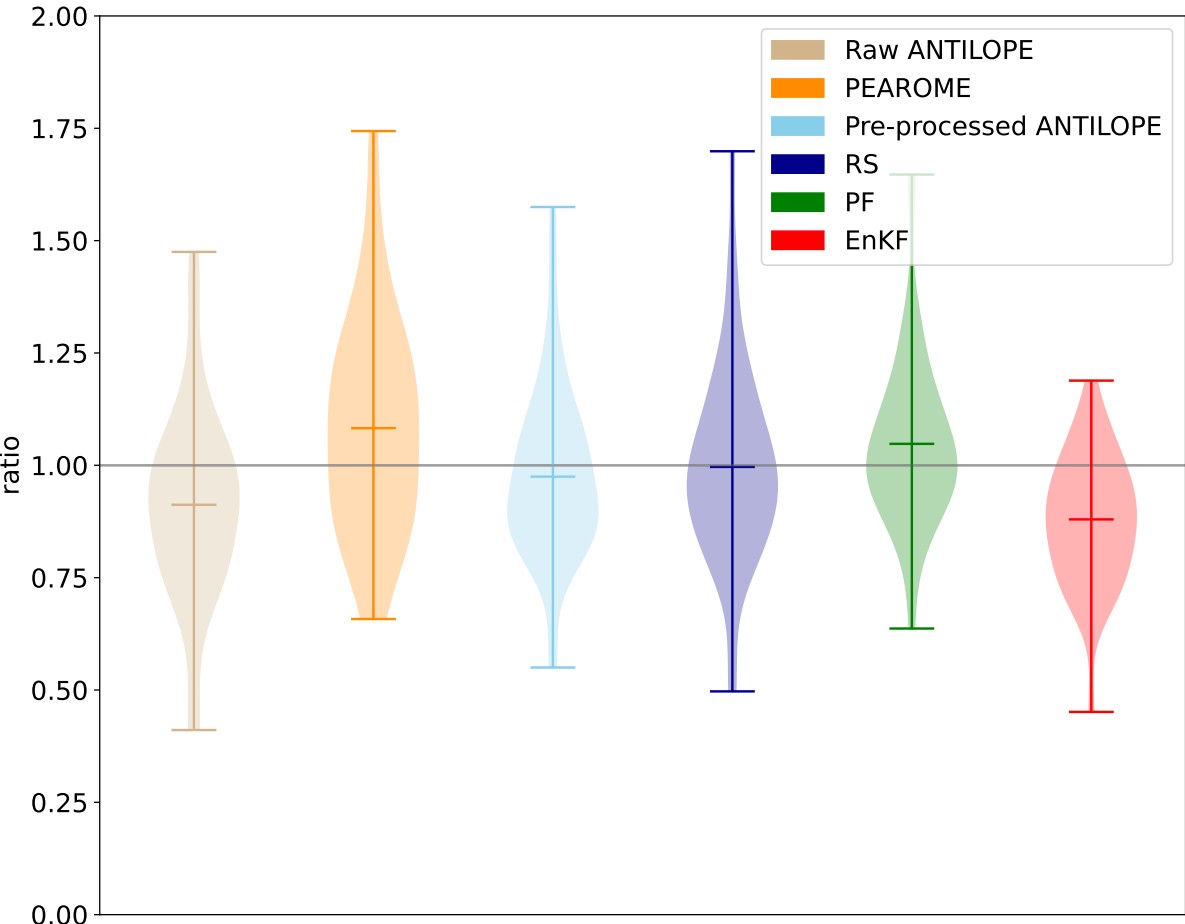

**Figure 8.** Distribution of the ratio between the precipitation accumulation estimated by the different products and the ski-resorts reference observations.

### 4.2.3 Spread-skill consistency

The spread of the raw PEAROME ensemble and the three ensemble analyses is shown in Figure 10 as a function of the ensemble mean. Ideally, the spread should be of the same magnitude as the quadratic error of the ensemble mean (along the black line Hopson, 2014). The post-processing of the PEAROME ensemble (Figure 10a) is specifically designed to optimise statistical ensemble properties such as the mean spread. Consistently, the magnitude of the ensemble spread is comparable to the magnitude of the ensemble mean error, with approximately the same mean value (indicated by the red dashed lines).

The spread skill of the RS analysis (Figure 10b) shows that the magnitude of the ensemble spread is also quite comparable to the magnitude of the ensemble mean error, with the spread on average slightly higher than the error (red dashed lines). The blue regression line shows that high errors also tend to be underestimated by the ensemble spread, but to a lesser extent. The





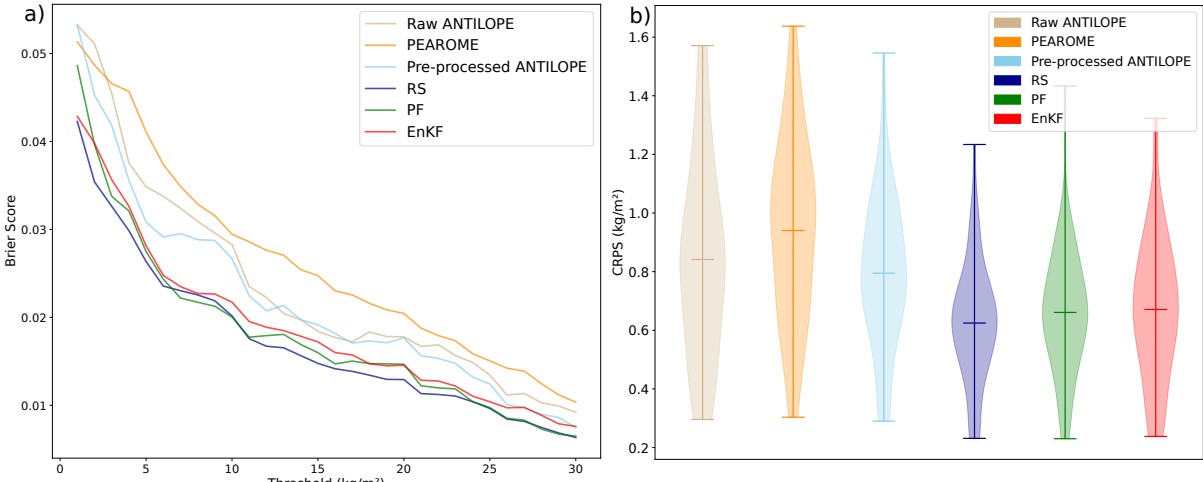

**Figure 9.** Evolution of the mean Brier score over all evaluation stations and dates for different daily precipitation thresholds ranging from $1 \mathrm{~kg~m}^{-2}$ to $30 \mathrm{~kg~m}^{-2}$ (a). Distribution of the mean continuous rank probability score over the reference stations (b). Six precipitation estimation products are shown: existing products in orange shades, deterministic products in light colors and ensemble products in dark colors.

RS ensemble spread is more often close to $0 \mathrm{~kg~m}^{-2}$, indicating a higher confidence in the precipitation estimate than the PEAROME ensemble.

Figures 10c and d show that the spread of the two ensemble analyses based on ensemble data assimilation methods clearly

underestimate the magnitude of the ensemble mean error. The spread of both analyses barely exceeds $10 \mathrm{~kg~m}^{-2}$, while the ensemble mean error is often above $15 \mathrm{~kg~m}^{-2}$.

It is also worth noting that the ensemble mean error of the three ensemble analyses is on average significantly lower than that of the PEAROME ensemble (vertical red dashed lines indicate an error almost twice lower) and does not reach the same extreme values (in particular, the RS analysis mean error exceeds $25 \mathrm{~kg~m}^{-2}$ only once).

### 4.2.4    Spatial patterns

Finally, Figure 11 shows an example of four precipitation fields for each of the four ensemble products over the Mont-Blanc area (see Figure 1) for the same date as the fields in Figure 5. This illustrates the spatial variability of precipitation produced by the different analysis methods. The RS analysis produces very smooth fields, similar to the corresponding preprocessed ANTILOPE field (Figure 5e), but with different precipitation intensities. The fields resulting from the PF and EnKF analyses

are a combination of this pre-processed ANTILOPE field and the corresponding PEAROME fields, also visible in Figure 11. Despite the field reconstruction step described in section 3.2.2, the application of the particle filter algorithm at the pixel scale results in rather noisy fields. EnKF precipitation fields better preserve the spatial variability of the corresponding PEAROME fields, with a convergence of intensity towards the pre-processed ANTILOPE field. As explained in section 3.3, the lack of





**Figure 10.** Spread skill of the different ensemble products. Each cross represents a specific evaluation point and date, its color indicates the corresponding 24 h precipitation. The blue line is the regression line of the scatter plot and the horizontal (resp. vertical) red line shows the mean ensemble error (resp. ensemble spread).





**Figure 11.** Example of 24 h precipitation fields of 4 December 2021 over the Mont-Blanc area (see Figure 1). a) Raw ANTILOPE observation. b) Pre-processed ANTILOPE field. c) Members 1 to 4 of the post-processed PEAROME ensemble. d) Members 1 to 4 of the Random Sampling analysis around the pre-processed ANTILOPE field (b). e) Members 1 to 4 of the Particle Filter analysis. f) Members 1 to 4 of the Ensemble Kalman Filter analysis. The full 16-member ensembles for each ensemble product are shown in Appendix B.

spatialised reference observations prevents any objective evaluation of the spatial structures of the different analyses, except

that noisy PF precipitation fields are a serious shortcoming for further exploitation of this precipitation analysis.





## 5 Discussion

### 5.1 Precipitation estimation products in mountainous areas

The evaluation of different precipitation estimation products with an independent observation network in ski resorts (section 4.1) shows that observation-based estimates are more competitive than forecasts from the PE-AROME ensemble, even after
a post-processing step designed to remove biases and improve reliability. This contradicts, at least for the French Alps, the conclusion of Lundquist et al. (2019) that high-resolution precipitation models outperform observations in mountainous areas. However, these products have complementary qualities and shortcomings.

Although radar observation provides spatialised information, its quality is very heterogeneous in space and suffers from important limitations in mountainous areas. Ground clutter and partial beam blocking are common due to the complex topography,
and the management of these disturbances in the radar processing algorithm described in section 2.1 does not compensate for the loss of information.

The most obvious effect of the interception of a radar beam by a mountain is a significant underestimation of the mean precipitation above the pixels affected by ground clutter, which produces spatial artefacts in the precipitation fields. This degradation of precipitation estimation due to increased height of the lowest usable radar measurement above ground is particularly pro-
nounced for stratiform precipitation systems with relatively small vertical extent (**?**). This is the case for the majority of the winter precipitation events in the French Alps on which this study focuses, thus limiting the extrapolation of its conclusions to convective situations.

Radar-based precipitation estimates are also less reliable over valleys, where the height difference between the radar beams and the ground is more important (Figure 2) and precipitation variations below the beams are neglected. However, this problem
is compensated in the ANTILOPE product by the use of many in-situ measurements at these altitudes (Figure 1). On the contrary, higher elevation areas are under-sampled (Figure 1 shows that very few automatic gauges are available above 2000 m a.s.l.), so radar measurement problems at these altitudes cannot be well corrected. The lack of reference observations over ridges (Figure 1) also makes it impossible to assess the magnitude of the error at these altitudes. This led us to develop a preprocessing step, described in section 3.1.4, to mitigate the resulting spatial artefacts and provide a spatialised assessment of
the highly variable uncertainty of this product.

### 5.2 Observation error

Most of the results of this study are based on the spatialisation of the error associated with the ANTILOPE precipitation product. Despite the positive results at the local scale shown in section 4.2, this method suffers from some limitations and possible improvements have been identified :

– The WMA filter described in section 3.1.4 tends to smooth precipitation fields. This is a necessary compromise to remove unrealistic spatial patterns that would be very detrimental to snow modelling (with unrealistic snow amounts near ridges, as illustrated by Haddjeri et al., 2023, with the ANTILOPE raw product). However, it can potentially erase realistic





spatial structures well detected by the radar, even if the formulation of the equation 11 and the parameters in the error estimation method have been chosen to ensure minimal modification over pixels with no clear systematic errors. High intensity precipitation kernels are therefore often smoothed by the preprocessing step, which may affect the detection of extreme events. The spatial structures of the analysed precipitation fields could not be evaluated with the reference data set used in this study. An indirect evaluation of simulated snow depths based on the precipitation analyses presented here is planned in a future study, following the workflow already proposed by Haddjeri et al. (2023). It will compare simulated snow depths with snow depth maps derived from high resolution Pleiades satellite images (Deschamps-Berger et al., 2020) over the Grandes Rousses domain.

– The climatological approach of this method makes it highly dependent on ANTILOPE evolutions (changes in the algorithm, in the available radar or in the in-situ measurements used), so that regular recalibration would probably be required.

– Additional available radar information, not considered in this work, could also provide relevant information for the estimation of the observation error (height above ground of the lowest available radar beam, position of ground clutter,...).

– As most gauges are located in valleys or at low altitudes, the uncertainty index is generally higher at high altitudes (Figure 4b). To mitigate the over-representation of low elevation pixels in the WMA filter, the weights computed in the preprocessing step (section 3.1.4) could take into account the elevation difference between pixels.

## 5.3 Choice of the data assimilation method

The random sampling ensemble analysis (RS) combined with the preprocessing method described in section 3.1.4 proved to fulfil most of the requirements for a suitable ensemble precipitation analysis in complex terrain, even if the accuracy of the spatial structures of the individual fields remains to be evaluated. This result is particularly promising for the future goal of assimilating satellite snow observations into snowpack simulations. It accurately accounts for precipitation uncertainties and thus addresses the main limitation raised by Cluzet et al. (2022); Deschamps-Berger et al. (2022) to take full advantage of the assimilation of snow observations. It is based on a very simple ensemble generator and could probably be significantly improved using methods inspired by those reviewed in Mandapaka and Germann (2010). The RS analysis is also the most computationally efficient and requires less data handling.

On the contrary, the information provided by high-resolution NWP models may suffer from larger errors on average, but is more physically consistent and can therefore be expected to have more spatially homogeneous errors. Ensemble forecasts also provide uncertainty information that is lacking in deterministic observational products. This motivated the use of the AROME climatology for the ANTILOPE pre-processing step, as well as the PEAROME daily precipitation fields through data assimilation methods. However, the evaluation presented in section 4.2 shows that most of the NWP added value comes from the use of the AROME climatology, at least at the local scale. In particular, the EnKF and PF analyses proved to be very under-dispersive. This deficiency could be very detrimental in snow simulation systems based on ensemble assimilation of snow observations with e.g. the particle filter Cluzet et al. (2022); Deschamps-Berger et al. (2022), as in many cases no optimal scenario would be



available among the backgrounds. The assimilation of snow observations would then fail to provide better estimates of snow conditions. The under-dispersion of both PF and EnKF analyses is an inherent aspect of these ensemble data assimilation algorithms: the main concept of these algorithms is to modify an ensemble of initial simulations (the background state) with the knowledge of an observation and its associated error. Both PF and EnKF methods ensure that the resulting analysed ensemble
has a smaller dispersion than the background ensemble: the PF analysis is a sub-ensemble of the background ensemble and the EnKF equations described in section 3.2.3 imply a convergence of all background members to the observation as long as the observation error is comparable to the background one (Hotta and Ota, 2021)). Thus, a well-dispersed background ensemble, such as the post-processed PEAROME ensemble used in this study, can only lead to under-disperse analysed ensembles when PF or EnKF ensemble data assimilation algorithms are applied using observations with lower error than the background, such
as the ANTILOPE product.

The consistency of the spatial structures produced by these analyses could not be objectively assessed. However, the PF analysis fields suffer from obvious spatial artefacts (even after a field reconstruction step). This PF method is also numerically more expensive when applied at the pixel scale, which can be a severe limitation for application over large areas. Applying the PF algorithm simultaneously to the whole domain would lead to degeneracy due to the very high number of available
observations compared to the ensemble size. A compromise solution would be to apply the PF algorithm over well-chosen subdomains, as suggested by Cluzet et al. (2021), but this would require further strong hypotheses for the definition of the subdomains. Considering all these elements, the analysis method based on the EnKF algorithm is clearly more suitable than the one based on the PF, but it does not outperform the RS analysis based on observations only and relying on a simplistic ensemble generation method, at least at the local scale used for the evaluation in this study. This result implies that the information
provided by direct radar and in-situ precipitation measurements cannot be significantly improved by daily high-resolution NWP forecasts up to 33 h lead time, despite major shortcomings. However, only a spatialised evaluation could determine whether the spatial structures produced by a high-resolution NWP model and transferred to the EnKF analysis (see Figure 11) can prove to be complementary information with added value.

## 6  Conclusions

This study first provides an evaluation of the advanced precipitation estimation products available over the French Alps. It focuses in particular on radar-based observational products, the quality of which is not well documented, but is often considered to be poor in complex terrain. Our evaluation against an independent dataset of local 24 h precipitation from ski resorts showed that radar measurements combined with precipitation estimation from an automatic gauge network provide the best precipitation estimates among all products considered. In particular, the performance of such a product was shown to be significantly
better than the output of the most advanced high-resolution numerical weather prediction model. This is noteworthy because previous studies have suggested that these models have outperformed observation-based precipitation estimates in some areas (Lundquist et al., 2019). However, significant shortcomings related to expected radar measurement problems were also identified. The authors developed a method to estimate the spatial distribution of climatological errors associated with unrealistic



spatial patterns, such as unrealistically low accumulation over ridges, in annual and daily precipitation fields. This method has
been developed and evaluated for the French Alps, but can be applied to any mountainous area covered by similar precipitation
estimation products. They also applied a correction algorithm to daily precipitation estimates using the mean precipitation
gradient from the AROME NWP model. Although no objective evaluation of the spatial structure of the daily precipitation
fields produced by this method could be carried out, this correction seems to be able to mitigate spatial artefacts visible in the
annual precipitation accumulations and proved to improve precipitation estimates at the local scale. Some possible refinements
of the method have also been outlined for future improvements.

Three different ensemble analysis methods were then implemented based on the corrected precipitation fields obtained by
the previous method. A first reference analysis randomly samples values around these observed precipitation fields from their
estimated error. Analyses were then conducted using two ensemble data assimilation algorithms, namely the particle filter and
the ensemble Kalman filter. These algorithms were used to combine the corrected precipitation fields with those produced by
a high-resolution numerical weather prediction model ensemble. The aim was to explore the complementarity of these two
sources of information. The method based on the particle filter turned out to be both more numerically expensive and to suffer
from major drawbacks, especially regarding the excessive noise in the produced precipitation fields. The method based on the
ensemble Kalman filter proved to be more suitable, but did not outperform the ensemble analysis based only on the reference
observation dataset used in this study. However, the spatial structures obtained by both methods are significantly different and
a spatialised evaluation will be necessary to determine which is the most appropriate.

Therefore, the next step will be to use the ensemble precipitation analyses described in this article to force a snow model.
This will allow the resulting simulated snow properties to be compared with corresponding satellite observations thanks to
the memory effect of the snowpack, which ensures a strong relationship between the state of the snowpack and past snowfall
(Haddjeri et al., 2023). The main challenge of this indirect evaluation of precipitation analyses will be the introduction of
additional uncertainties such as the height of the rain-snow limit (Vionnet et al., 2022) and the errors of the snowpack model
(Essery et al., 2013; Lafaysse et al., 2017).

*Code and data availability.* All data used in this study are open access and can be downloaded online : https://portail-api.meteofrance.fr/
web/fr/. The various codes used to process the data, perform evaluations, apply the correction method and perfrom ensemble analyses are
available on request at https://github.com/vernaym/These.

*Author contributions.* M.V collected all the data, carried out the various simulations and evaluations, developed the method and wrote the
paper. M.L and C.A supervised every stage of this work, directed various scientific decisions and contributed to the writing of this paper.

*Competing interests.* No competing interests where involved in this paper.



*Acknowledgements.* This work was partly funded by the TRISHNA-Cryosphere project of the Centre National d'Etudes Spatiales (CNES) and the SENSASS project funded by the Region Auvergne Rhône-Alpes (France). The CNRM-CEN is part of LabEx OSUG@2020. The authors thank Dominique Faure and the Meteo France radar team for their help in understanding radar measurement issues in mountainous areas.



## Appendix A: Evaluation scores

The Brier Score (BS) assesses the ability of an ensemble to forecast a threshold being exceeded. For each given event k, the predicted probability $y_k$ of a given threshold being exceeded (the number of members forecasting the event divided by the size of the ensemble) is compared with the corresponding binary observation $o_k$ of the threshold being exceeded, and for N events the BS is given by :

$$BS = \frac{1}{N} \sum_{k=1}^{N} (y_k - o_k)^2 \tag{A1}$$

Although the BS has was originally designed for ensemble systems, it is also commonly used in a deterministic context DeMaria et al. (2009); Vernay et al. (2015), where the predicted probability $y_k$ only takes values of 0 and 1. The Brier Score ranges from 0 to 1, with 0 corresponding to a perfect score. It is computed for different precipitation thresholds from $1 \, \mathrm{kg \, m^{-2}}$ to $30 \, \mathrm{kg \, m^{-2}}$ to check the ability of each product to forecast a variety of precipitation events.

The Continuous Rank Probability Score (CRPS, Matheson and Winkler (1976)) is a measure of the difference between the ensemble cumulative distribution function $F_y$ predicted for a given event and the Heaviside function centered on the associated observation $H_o$ :

$$CRPS = \int_{\mathbb{R}} (F_y(z) - H_o(z))^2 dz \tag{A2}$$

Unlike to the ensemble mean bias, the CRPS takes into account the error of each member of the ensemble: an unbiased ensemble with a large spread can have a higher CRPS than a slightly biased ensemble with a small spread.



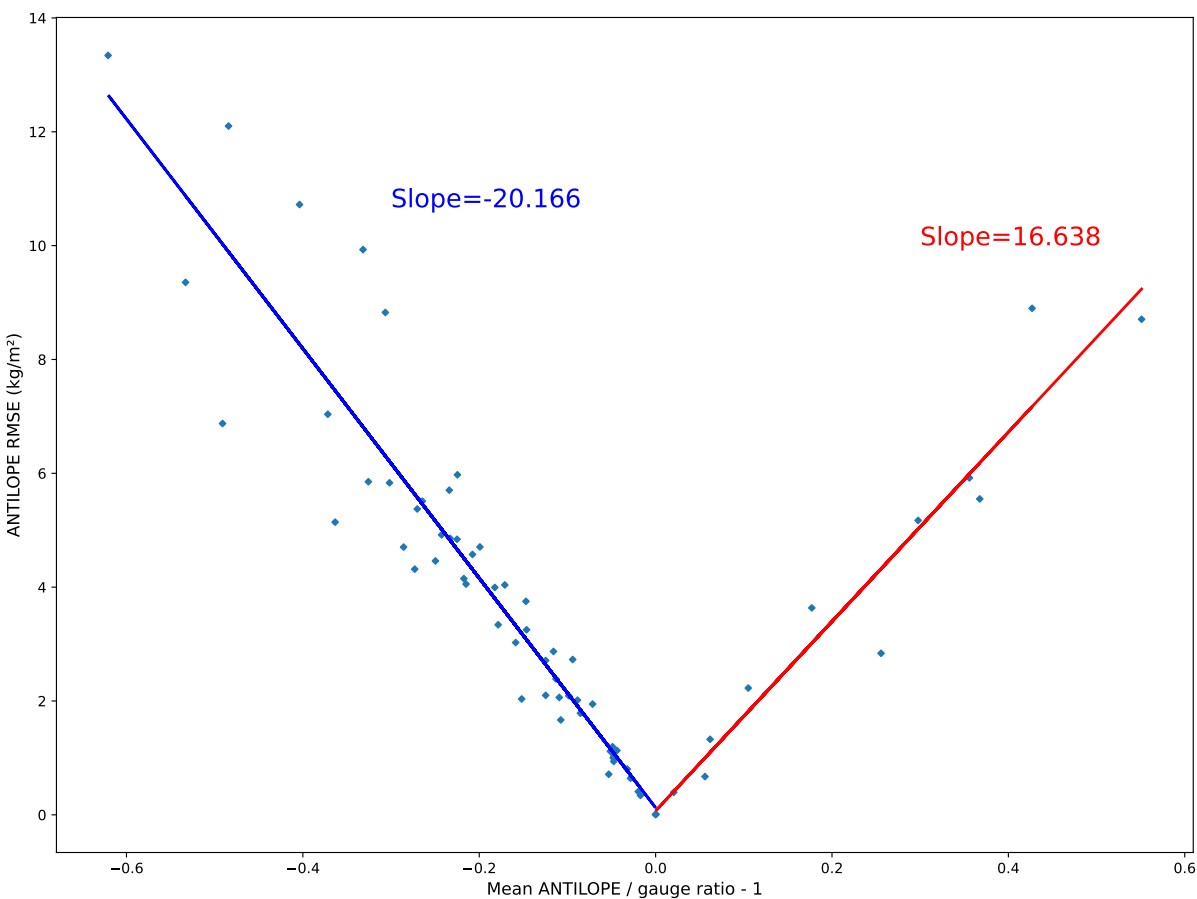

**Figure A1.** Linear regression between ANTILOPE / gauge mean ratio ($R^{\text{GAUGE}} - 1$) and the corresponding rmse over the 68 reference stations. The slope coefficients are used in the method described in Section 3.1.1

## Appendix B: Additional results



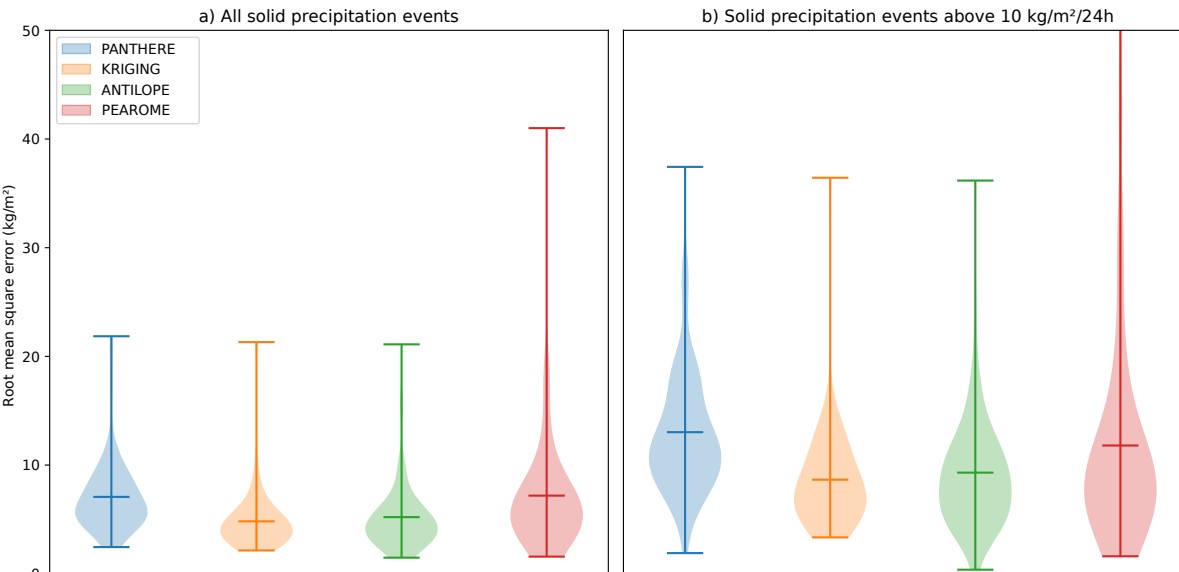

**Figure B1.** Same as Figure 6 but considering only snowfall events. The precipitation phase is determined by the additional ski-resort observation of the maximum altitude reached by rain during the observation period. Only situations where this maximum altitude is below the station altitude are considered here.





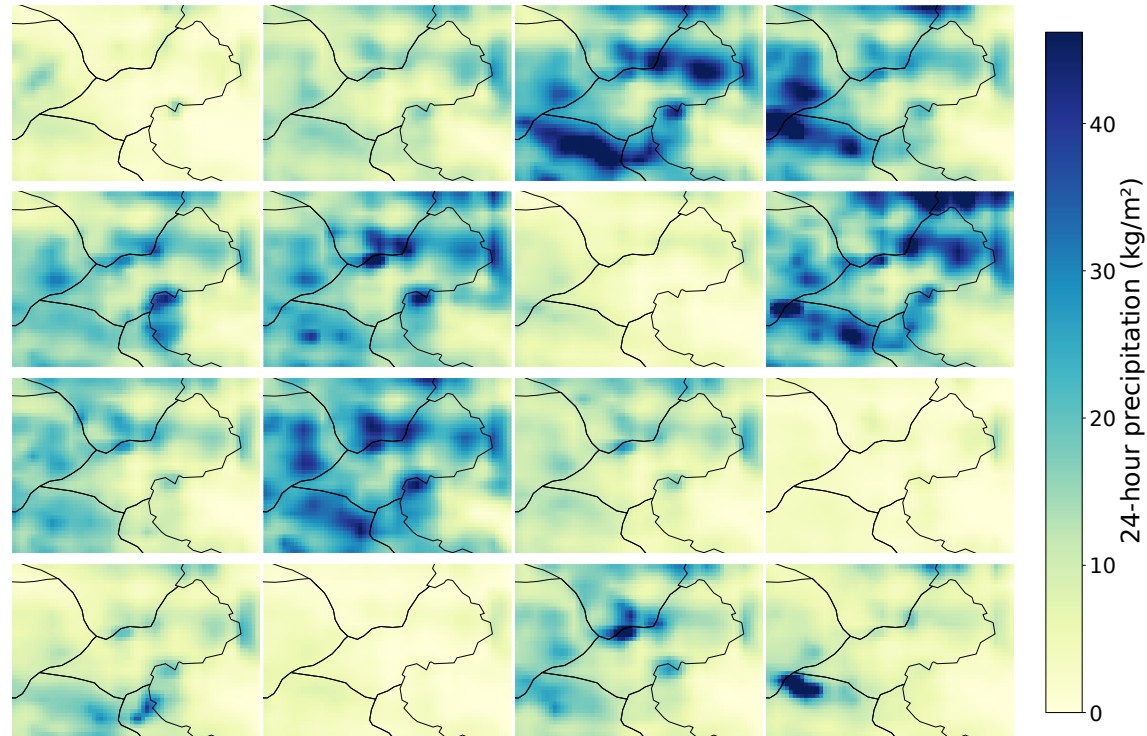

**Figure D1.** Raw post-processed PEAROME ensemble 24 h precipitation fields of 4 December 2021, downscaled to the ANTILOPE 1 km resolution over the Mont-Blanc area (see Figure 1). The corresponding raw ANTILOPE observation and the associated pre-processed fields are shown in Figure 11.

**Appendix C: Ensemble precipitation analyses**



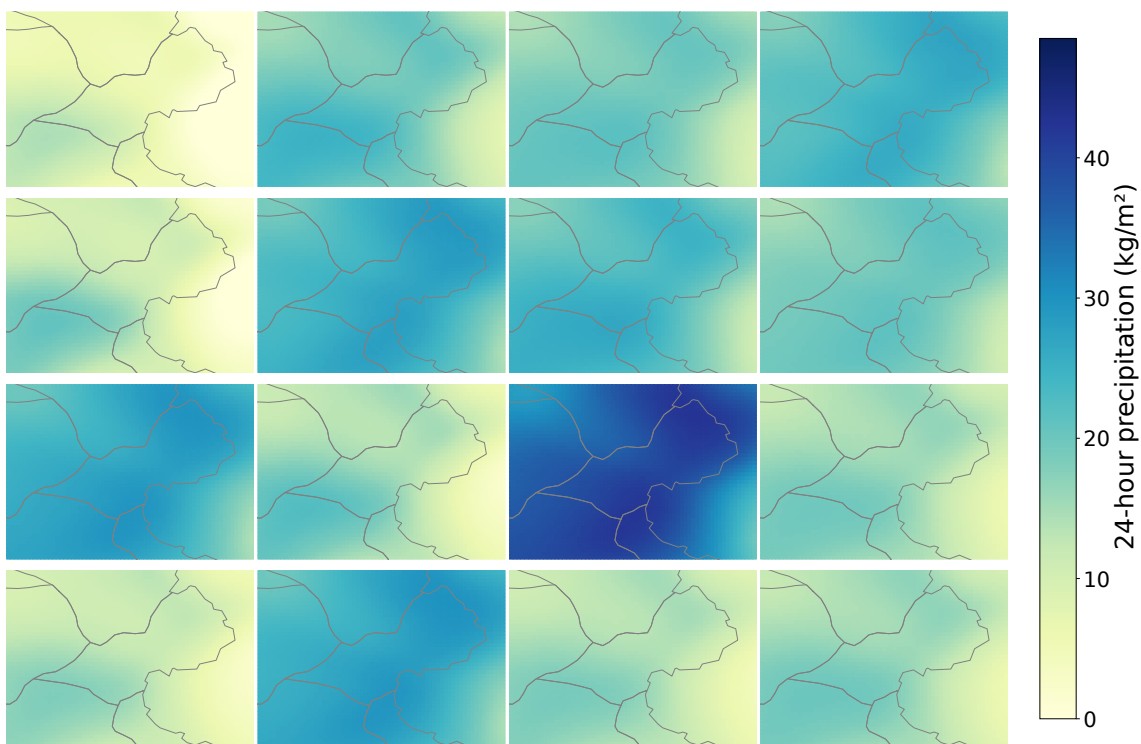

**Figure E1.** 24 h ensemble precipitation analysis of 4 December 2021 over the Mont-Blanc area (see Figure 1) obtained by a random sampling around the pre-processed ANTILOPE field (Section 3.2.1). The corresponding raw ANTILOPE observation and the associated pre-processed fields are shown in Figure 11.

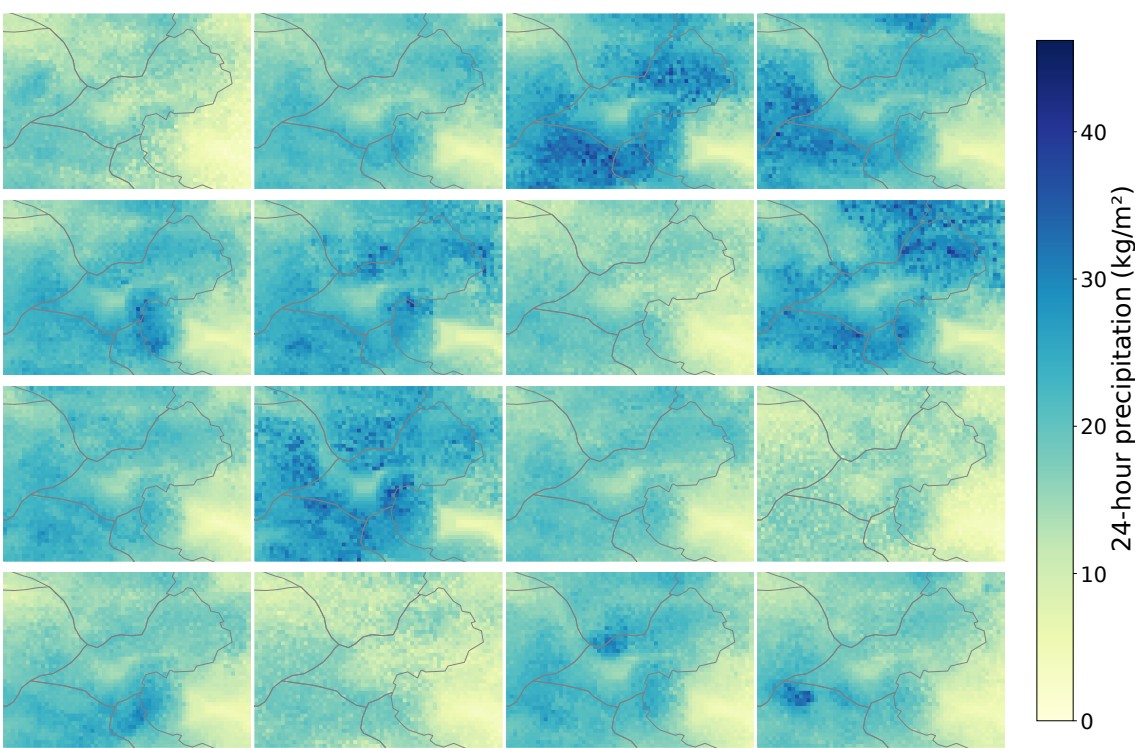

**Figure F1.** 24 h ensemble precipitation analysis of 4 December 2021 over the Mont-Blanc area (see Figure 1) obtained with the Particle Filter method (Section 3.2.2). The corresponding raw ANTILOPE observation and the associated pre-processed fields are shown in Figure 11 and the post-processed PEAROME ensemble is shown in Figure D1.

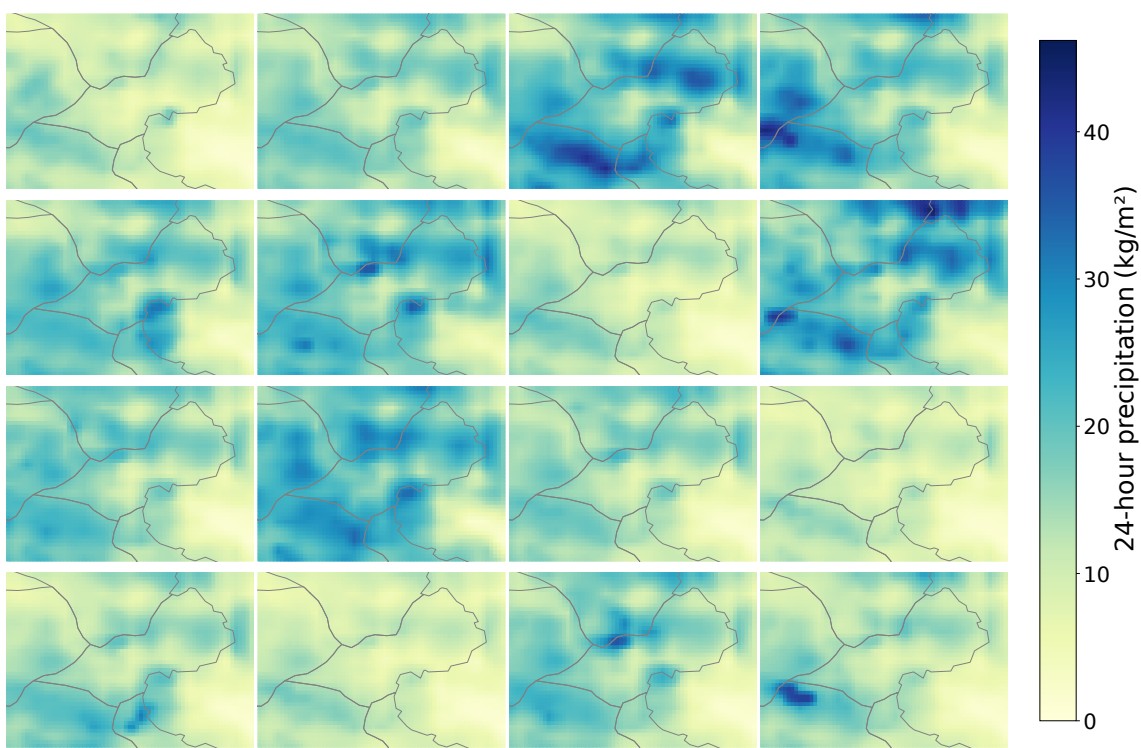

**Figure G1.** 24 h ensemble precipitation analysis of 4 December 2021 over the Mont-Blanc area (see Figure 1) obtained with the Ensemble Kalman Filter method (Section 3.2.3). The corresponding raw ANTILOPE observation and the associated pre-processed fields are shown in Figure 11 and the post-processed PEAROME ensemble is shown in Figure D1.



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
