# Peer review of "Radar based high resolution ensemble precipitation analyses over the French Alps"

_EGUsphere, 2024_

## Author Comment (AC1)

**Specific comments**

→ "Their" refers to "precipitation fields", the phrase has been modified to make it more explicit.

Lines 6-9: The combination of the two sentences seems to suggest that the radar product is unbiased, in contrast to the precipitation fields from the biased NWP products. Is this correct?

Radar products can be locally biased (among other shortcomings) while NWP products suffer from widespread systematic biases. This precision has been added in lines 6-9.

Lines 13-14: Please rephrase "radar and gauges precipitation estimation products" to "precipitation estimates by radar and gauges" or similar.

The suggested rephrasing has been applied.

Line 19: Does "ski resort" provide any relevant information here?

Ski resorts sample typical high-stakes areas for all snow related activities. Therefore, specifying that the reference data come from ski resorts implies that the system is evaluated over areas where it is designed to be the most relevant.

Line 24: Maybe change "human activities" to "practical applications"?

The proposed modification has been applied

Lines 45-48: Please also mention that the precipitation measurements are affected by systematic undercatch in case of solid precipitation.

The issue of gauge undercatch in cause of solid precipitation has been added, thank you for pointing out this omission

Lines 58-78: These two paragraphs lack a clear statement about the research gap of current studies, and what research is needed for filling this gap.

These paragraphs have been adapted to more clearly identify the research gap of current studies and propose a way forward to address this gap. In particular, the following information have been added :

"The spatial structure of the error associated with [radar-based] products in mountainous areas and its overall magnitude have not been investigated in depth."

and

"[Existing ensemble precipitation analyses] were not designed to meet the requirements of snow data assimilation in a high-resolution snowpack modelling system."

Lines 80-83: Perhaps move these lines to the end of the paragraph.

Theses lines have been moved to the end of the paragraph as suggested.

Lines 90-91: Consider moving the sentence "Evaluation data is only available for the period from 1 December 2021 to 30 April 2022" to section 2.4 describing the evaluation data.

The proposed modification has been applied.

Line 90-94: Consider adding information here stating that 24 h sums of all precipitation products up to 08:00 CET were analyzed in this study (if this is the case).

This information has been added.

Section 2.4 and 2.5: The authors have chosen to first present grid-based products based on observations (section 2.1 to 2.3), followed by the point evaluation measurements (section 2.4), and finally the gridded data from the NWP (section 2.5). To me, it seems more logical to first present all gridded products (currently sections 2.1-2.3 and 2.5), followed by the evaluation data (currently section 2.4). Thus, consider swapping order of section 2.4 and 2.5.

Section 2.4 and 2.5 have been swapped as suggested, thank you for this very relevant change.

Figure 2: What is the blueish horizontal bar in the left panel?

This is the ANTILOPE precipitation accumulation along the cross section. The colorbar has been shifted in between the two sub-figures and an explanation added to the legend to make it more explicit.

Figure 2: The text in the figure is very small. Please try to enlarge where possible.

Font size has been enlarge.

Figure 2, caption: Consider changing the color of the line in Figure 1 showing the radar beam from Pic Blanc to Grandes Rousses massif from gray to a more prominent color.

The colour of the line in Figure 1 indicating the cross section of Figure 2 has been modified to purple.

Heading 2.2: Consider changing to "radar-rain gauge combination" which seems to be a more common terminology (Ochoa-Rodriguez et al., 2019).

The proposed modification has been applied

Line 130: I wonder whether the usage of "1x1 km^2" is correct. Maybe "1 by 1 km" is better. Or even "…is available at a 1 km resolution…". If adapting another convention, please change throughout the paper.

The convention "... X km resolution…" has been adopted throughout the paper

Line 139: Rephrase from "the errors' magnitude" to "the magnitude of the errors".

The phase has been modified, thank you for this correction

Section 2.4: Please provide the number of stations including their minimum, mean and maximum altitude.

These informations have been added in section 2.4, as well as the number of stations above 2000m a.s.l in section 5.1 (line 482)

Lines 159-162: Add a reference to a study about undercatch issues with precipitation gauges such as Rasmussen et al. (2012).

References to Rasmussen et al. (2012) and Kochendorfer et al. (2020) have been added

Line 160: Change from "under-catchment" to "undercatch".

The change has been applied

Lines 167-170: Why were precipitation sums calculated as a 24 h sum up to 06:00 UTC and not up to 08:00 CET as with the other products? What influence does this have on the analysis presented in the study?

Precipitation were indeed summed up to 07:UTC (8:00 CET winter time) as with the other products. This mistake has been fixed, thank you for pointing it out. However, few sentences have been added in section 2.5 to discuss the (limited) impact of time change on the analysis in this study (lines 181 to 185). This gap is expected to have very limited impact on the results, as fewer reference observations are available in April (due to the closure of ski resorts) and only two significant precipitation events occur during this period

Line 177: What region does "metropolitan France" refer to?

Metropolitan France refers to mainland France, the world has been modified

Line 188: Change from "half those" to "half of those".

The correction has been made, thanks for noticing

Line 188: Change from "in between" to "between the two peaks".

The rewording has been applied.

Lines 188-190: Please rephrase this sentence since it currently does not read well. Maybe even spilt into two sentences.

The sentence has been split in 2 as proposed.

Lines 205-206: Please use either "true" or "real" precipitation, and align this with the notation used for the variables. Likely, "true" is a better terminology than "real" precipitation.

The world "true" has been applied everywhere, thank you for noticing this inconsistency.

Equation 1: Consider renaming the variable representing accumulated precipitation (RR) to an acronym that cannot be confused with the variable representing the ratios (R). The current usage of variable names is confusing.

The variable representing precipitation has been changed from "RR" to "P"

Line 215: Change from "gauges locations" to "location of gauges".

The proposed modification has been applied

Line 222: Change from "far from pixel i of a distance d_ik" to "with a distance d_ik separating point i and k".

The rewording has been applied

Line 225: What does this ratio express? Please provide a meaning to the reader for easier understanding of the methods.

The following explanation has been added (line 270):

"This estimated ratio conveys the hypothesis that the expected precipitation accumulation over an unobserved pixel can be retrieved from the precipitation accumulation observed at a nearby gauge

by applying the AROME precipitation accumulation ratio between these two locations. The underlying hypothesis is that AROME simulates realistic vertical gradients of precipitation even if it may be biased"

Section 3.1.1 to 3.1.4: The explanation of these methods needs to be improved. Potentially a hypothetical numerical example will help the reader to understand the methods better illustrating the core concept of the method using (a) one location including a gauge observation, (b) one location where the ANTILOPE is free of artifacts, and (c) a final location where ANTILOPE is affected by ground cluttering.

A supplementary appendix has been included to provide illustrative numerical examples of the core concept of the method as suggested ("Appendix A :  ANTILOPE climatological error estimation method illustration", page 29) and some clarifications have been made in sections 3.1.1 to 3.1.4. Additionally, Figure 5 has also been revised to better illustrate the impact of the different steps of the method.

Lines 291-306: What do the authors mean with a spatial observation error and a dynamic error? Is the first constant in space while the second varies in space?

The spatial observation error refers to the spatial structure of the error associated with measurement issues (variable in space) and the dynamic error refers to the uncertainties associated with each individual precipitation event (variable in time). These precisions have been added in the text (lines 318-320):

"The spatial observation error $E_i$ accounts for the intrinsic spatial structure of the error associated with measurement

issues in the ANTILOPE product described in section 3.1 and comes from the preprocessing step (Section 3.1.4). The dynamic

error is associated to the uncertainty of each individual event and is expressed as a fraction of the precipitation intensity."

Lines 321-322: Reformulate. Likely the reference does suggest to reduce the number of assimilated observations in general, but not to reduce the number of particles to 16 and the number of observations to 1 in particular, as I understand the current text.

The explanation has been modified as suggested (line 340) :

"To deal with the known problems of the PF algorithm for large numbers of observations, Snyder et al. (2008) suggests to reduce the dimension of the problem by splitting a large set of observation into smaller subsets. Here we chose to apply the particle filter algorithm for each pixel independently, reducing the problem dimension to 1 observation for a 16-member ensemble."

Line 353: What does "improved predictive added value" mean?

This part has been reworded to avoid the use of this unclear formulation, without using bullet points as suggested in another remark (lines 375-377) :

"This ensemble analysis should reduce the systematic biases of the ensemble mean and improve precipitation estimation as compared to other existing products. Finally, spatial artefacts affecting

the radar-based precipitation fields must not be propagated to the analysis, meaning that each individual ensemble member should feature realistic spatial structures"

Line 355: Consider removing "local" from the sentence.

"Local scale" has been replaced by "point scale"

Line 361: What is the difference between systematic errors and biases?

The wording of the phrase has been modified to remove the confusion (lines 384-385):

 In the case of deterministic products, this evaluation focuses on the magnitude of systematic biases and their spatial distribution as well as on the consistency of the spatial structure of the precipitation fields.

Lines 367-368: Change from "the estimated and observed 24 h precipitation values time series" to "the estimated and observed time series of 24 h precipitation records" or similar.

The proposed rewording has been applied

Line 386: Repetition of "when only solid precipitation is considered".

Thank you for noticing, the duplicate has ben removed

Line 387-390: Consider swapping order of these two sentences.

The two sentences have been swapped as suggested

The reference to figure 9 showing those results has been added

Line 402: The first sentence on this line should probably be combined with the previous sentence.

The two sentences have been combined (lines 424-425):

"The ensemble precipitation analyses in this study are based on the ANTILOPE preprocessing step described in Section 3.1.4, which relies on the error estimation method described in Section 3.1.1."

Section 4: For the reader, it is likely easier if single sections presents results with a focus on individual figures. As an example, results from Figure 8 and 9 are presented in section 4.2.1 on lines 407-409, and again in section 4.2.2 on lines 411-417. In this example, the heading of the first section could be changed to "Impact of pre-processing procedure on spatial precipitation patterns" and the second section to "Skill of the precipitation estimates" (or similar) to allow for a focus of individual figures in single sections. Of course, cross references are allowed, but the current presentation of the results is confusing to me.

The position of the figures where fixed to respect your recommendation as best as possible. Only Figures 7 and 8 remain associated to the same section.

Figure 8: Please find a more descriptive label for the vertical axis.

The vertical axis label has been changed to "Precipitation estimation / reference ratio"

Figure 9, caption: Rephrase to "Ratio between estimated and observed accumulated precipitation for the different methods assessed in this study" or similar.

The caption of Figure **8** has be changed to:

"Ratio between estimated and observed accumulated precipitation over all ski-resorts reference stations for the different methods assessed in this study"

Lines 420-421: Change from "(along the black line Hopson, 2014)" to "(along the one-to-one line) as described by Hopson et al. (2014)" or similar.

The proposed modification has been applied

Line 424: What does "quite comparable" mean? Please avoid fuzzy terms.

A more precise rewording has been applied (lines 450-451) :

The spread skill of the RS analysis (Figure 10b) shows that the magnitude of the ensemble spread match the magnitude of the ensemble mean error, with the spread on average slightly higher than the error (red dashed lines).

Line 426: "To a lesser extent" than what?

A rephrasing has been applied to clarify the meaning.

Figure 10, caption: Please split the last sentence into two sentences to improve readability.

The last sentence has been split in two as requested

Line 448: Remove "estimation" in the part "precipitation estimation products".

The word "estimation" has been removed

Line 449: Consider changing from "more competitive" to "provide better results".

The proposed modification has been applied

Line 465-467: Please provide the number of stations above 2000 m.a.s.l.

The number of stations above 2000m (31) has been added line 465-467 (now 473-474) , as well as the total number of stations (512)

Line 473: Consider changing from "suffers from some" to "has".

The proposed change has been applied

Line 483: Consider changing from "It will compare" to "In such a study, we will".

The rewording has been applied

Lines 475-493: In my opinion, the authors overuse bullet list making the text difficult to read. Please consider reducing the use of bullet points throughout the paper.

The author tried to reduce then use of bullet list as suggested.

Line 504: "More spatially homogeneous errors" than what and why?

Thank you for pointing out this inaccuracy, the sentence has been completed.

Line 519-520: Note that a precipitation product with shortcomings was assimilated and not direct observations. Please discuss the implications of this approach for the final results.

The following precisions have been added (lines 544-546) :

"This is typically the case with the ANTILOPE product, at least where reference data is available. However, in instances where the ANTILOPE uncertainty is significantly higher than the PEAROME error, the impact of these algorithms on the background ensemble is minimal."

Line 531: What does "despite major shortcomings" refer to?

It refers to spatial artefacts in the radar based precipitation product. However, the point of the last sentence is to specify that we do not know if these spatial artefacts have been mitigated in a satisfactory manner so this precision has been removed.

Line 535: What does "the advanced" mean in this context?

The term implies that the best available products are considered. It has been replaced by "state of the art" for clarification.

Line 537: Consider changing from "local 24 h precipitation" to "24 h precipitation sums" or similar.

"local 24 h precipitation" has been replaced by "24 h precipitation accumulation"

Line 540: Is the ARMOE system "the most advanced high-resolution numerical weather prediction model" of all NWPs in the world?

PEAROME is the only high resolution ensemble model available over the study area of the French Alps, this precision has been added.

Line 543: Consider changing from "The authors" to "In this study, we".

The proposed modification has been applied

Line 546: Reformulate. Change from "They also applied a correction algorithm" to "A correction algorithm was applied" or similar.

The proposed rewording has been applied

Line 549: "Local" seems to refer to station observations. Please clarify.

The phrase has been modified to clarify that the proven improvement only concerns the set of reference stations.

Line 550: Please give one or two examples of possible refinements in the conclusion.

Two examples of possible refinements presented in section 5.2 have been added

**Technical comments**

Line 37: A misplaced comma.

The comma has been moved to the right place

Line 44: Rephrase to "precipitation inputs provided".

The rephrasing has been applied

Line 59-61: There seems to be an error with the usage of parenthesis here.

The parenthesis usage has been fixed

Line 80: Remove "preliminary".

The modification has been done

Line 148: Change from "Fig. 1" to "Figure 1".

The change has been made

Line 379: Wrong units: "g m^-2".

The unit has been fixed, thank you for pointing out this mistake

Line 380: "This Figure". Misplace capital letter.

The capital letter has been removed

Line 383: Inconsistent usage of units: "kg/m2/24h". Please also correct many of the figure titles.

Figures 3, 6, 7, 9, 10, 11, A1, B1, C1, D1, E1, F1, G1 have been updated

Line 387: Likely refers to appendix B and not A.

The reference has been fixed

Line 393: A white space is missing.

White space added

Line 395: Error in figure reference.

Reference fixed

Line 401: Likely wrong reference to section 5.

Reference to section 3.1.4 fixed

Line 416: Change from "Figure 9" to "Figure 9b".

Change done

Line 440: Consider changing from "Figure 11" to "Figure 11e and f".

Sub-figures references added

Line 460: Wrong reference.

Reference fixed

Line 510: Wrong reference format.

Format fixed

Line 583: Remove "has" from the sentence.

Sentence fixed

---

## Author Comment (AC2)

One point that could be discussed, albeit briefly, is the uncertainty of the estimates obtained from the network of rain gauges, whether from the operational network or from the ski resorts. Like radar data, rain gauge data can be subject to measurement error, particularly in strong winds and/or snowy conditions. So, the different scores presented for the various radar products and rainfall ensembles take as their reference values from ground-based rain gauges, which can sometimes be accompanied by significant uncertainty. Another remark that is a little outside the scope of this article is that in general the kriging with external drift used to merge rainfall and radar data provides both an estimate of the rainfall values and also an estimate of the associated uncertainty that could be used. Perhaps this information could have been considered for this study.

--> What do the authors think?

The authors fully agree with the limitations of in-situ precipitation measurements as outlined in this remark. These limitations have already been discussed in the introduction and in the section "Evaluation data" (now section 2.5), and a mention of them has been added in the section "Gauge kriging) (now section 2.3) :

"As already mentioned, gauges measurements are known to be affected by significant undercatch in case of solid precipitation and in windy conditions (Rasmussen et al., 2012; Kochendorfer et al., 2020)"

However, in-situ observations from ski resorts are the only available independent data available for the evaluation of the various precipitation products. This precision has been added in Section 2.5 ("Evaluation data", line 197).

The under-catch issue in snowy conditions is also briefly discussed in section 4.1 ("Evaluation of existing precipitation products", line 407-410) :

"Similar results are obtained when only snowfall events are considered (Figure B1). Other results (frequency of relative errors less than 20%, not shown) also support the choice of ANTILOPE as the best precipitation product available for an ensemble analysis system, especially since ANTILOPE performs equally well when only solid precipitation is considered (see Figure B1 in Appendix C)."

The potential use of the uncertainty coming from the kriging with external drift algorithm has been added in the perspectives (Section 5.2, lines 521-522): "The uncertainty information generated by the kriging with external drift algorithm used in the ANTILOPE raw product could be used to improve the estimation of the associated observation error."
However, this uncertainty does not explicitly take into account the precipitation patterns associated with the relief, as does the AROME model, which is used in this study instead.

**Specific comments and questions**

Page 5: Figure 2 is a very good illustration of the problem encountered in mountainous areas. However, I'm not sure that the coloured band at 3500m is easy to understand and I wonder whether it could be misleading or misinterpreted. Is it an estimate of rainfall at a constant altitude of 3500 m or is it the cumulative total on the ground along the transect between the Moucherotte radar and the Pic Blanc? The chosen radar (Moucherotte in this case) is perhaps not the best one for this illustration, as there is a strong under-estimate near the Pic Blanc "upstream" / before the highest

summits, whereas these strong under-estimates should rather be the result of masking by the relief "downstream", i.e. after the relief. In all likelihood, this area of underestimation is more likely to be the result of the Colombis radar being masked by the Ecrins massif (between La Meije and Colombis radar)? In fact, what's hard to highlight from this coloured band is that it's a combination of data from several radars (or rather, at each point it's data from a single elevation selected from a single radar). Morover, the indication "radar information not used" and "use if corrected radar information" only apply here if only the Moucherotte radar was used.

-> Perhaps it would be better to remove this coloured band to avoid misinterpretations or misunderstandings?

The coloured band at 3500m is the yearly ANTILOPE precipitation accumulation along the transect between the Moucherotte radar and the Pic Blanc (the colorbar is common to both panels, it has been moved in between the two panels and the legend has been modified to improve the clarity of the figure). We think that it is important to show this coloured band because it reveals a major shortcoming of the ANTILOPE product that is not visible in observations. The main objective of this figure is to highlight the spatial artefacts affecting the ANTILOPE product and the need to mitigate them from a user's point of view.

Concerning the origin of the problem, a secondary objective of this figure is indeed to point out that areas with major precipitation under-estimation seem associated to ground clutter problems ("upstream" and above mountain ridges) rather than beam masking "downstream". This is supported by the fact that unrealistically low precipitation accumulations are observed "upstream" (because the bottom of the radar beam can reach the ground before the top) and above mountain ridges (within the area circle in red) and precipitation accumulations "downstream" (within the area circle in blue) are more consistent with precipitation in front of the relief.

We acknowledge the challenge of disentangling the sources of error when working with a composite product, but the use of such product is necessary for large scale applications. We have based our reasoning on the assumption that the contribution of the Moucherotte radar is dominant for the estimation of precipitation over the Pic Blanc since the Colombis radar is about twice as far away from the Pic Blanc (70km) as the Moucherotte radar (37km). Figure 1 below also shows that the quality index, which is used to weight the contribution of each radar to the final product for a given situation, is higher for the Moucherotte radar than for the Colombis radar in the vicinity of Pic Blanc.

[Figure]

Figure 1: Quality index of the Moucherotte radar QPE (left) and Colombis radar QPE (right) over the Grandes Rousses domain. This quality index is used to weight the contribution of each single radar QPE in the final composite QPE product.

Moreover, the Colombis radar beam towards the Pic Blanc only crosses the western part of the Ecrins massif with no peaks well above 3000m as shown in Figure 2 bellow. This figure also confirms that the underestimation of precipitation over the Pic blanc is not due to the masking of the Colombis radar beams, since the precipitation accumulation in the valley between the Ecrins massif and the Pic Blanc (between 50km and 60km from the Colombis radar and affected by the same masks) is much higher than that over the pic Blanc (the colour scale is the same as the one in Figure 2).

[Figure]

*Figure 2: Relief cross section between the Colombis radar and the Pic Blanc and corresponding yearly ANTILOPE precipitation accumulation.*

Page 5, Line 109: I propose to replace "measured" by "estimated"

The proposed modification has been applied.

Page 10, section 3.1.2: the ratios are estimated on an annual basis. ANTILOPE uses PANTHERE products, which combine data from different radars with elevation angles that can change over time depending on weather conditions and the availability of radars in real time.

-> Is this a problem and/or is this effect compensated/corrected in the dynamic stage of section 3.1.4?

The error estimation method is based on the assumption that the magnitude and spatial patterns of the ANTILOPE error remain relatively constant on average from year to year. This is a questionable assumption for the reasons that you have highlighted. The WMA step is only designed to remove spatial artefacts. It is not a reliable solution for cases where the ANTILOPE error deviates significantly from its climatology.

The following perspective has been added to address these issues (lines 523-524):

"The error estimation method described in Section 3.1.1 could be applied to daily precipitation fields in order to improve the handling of instances where the ANTILOPE error is significantly different from its climatological value."

Page 11, equation 8: Is it possible to give physical meanings to the two different terms in this equation?

The two terms in equation 8 can be seen as metrics of the two main sources of uncertainty associated with the method itself. The following information has been added in the explanation of equation 8 :

- [The first term] increases as the estimated ratio deviates from 1, thereby indicating a likely systematic bias.

- [The second term] increases with $S_i$, increasing the estimated uncertainty in case diverging ratio estimates are obtained when applying the method to different reference gauges.

Please note that a numerical illustration of the ANTILOPE error estimation method has been added ("Appendix A :  ANTILOPE climatological error estimation method illustration", page 29), which provides a practical interpretation of these two terms in different situations.

Page 12, line 273: About the weight cik: Can it tend towards infinity if Uk tends towards 0?

$U_k$ is indeed always greater than 2 : there was an error in the definition of $A_i$ and $B_i$, which are designed to be higher than 1. Equation 9 has been fixed (line 262), thank you for pointing out this mistake.

 Eq 10, Pk would be to be defined in the text

Equation 10 was a mistake, $P_k$ has been replaced by the proper value $R_{ik}^d$, thank you again for noticing this inconsistency.

Page 13: What exactly does Cii correspond to? I'm having trouble convincing myself that this could be a special case of Cik defined on the previous page.

$C_{ii}$ is the weight of pixel in in the window centred on pixel "i" as defined on the previous page. In this case $d_{ii}=0$ and $C_{ii}=1/U_i$ is inversely proportional to the estimated uncertainty of pixel "i".

Line 379: I propose to put kg/m2 instead of g/m2

The unit correction has been made, thanks for pointing it out

Line 399, title of 4.2: add the word "ensemble" in the title?

This section also deals with the evaluation of the deterministic ANTILOPE post-processing method, not only the ensemble analyses.

Line 255. Given the equation for U and the elements in Appendix 1, do you confirm that U have the unit kg/m²? If confirmed, I propose to mention it in the text and in the colorbar of fig. 5D

U is indeed in kg/m², this unit has been added in the text (line 259) as well as in the colorbar of fig. 5d and 4b.

Page 9: Figure 3a: what do the shapes used (triangles, circles) correspond to?

The following explanation has been added to the legend of Figure 3 and Figure 7 :

"The marker associated with each evaluation station is a circle if the ratio between ANTILOPE and the reference gauge precipitation estimates is not considered significant (between 0.8 and 1.2) and a triangle facing up (resp. down) if this ratio is significantly higher (resp. lower) than 1."

Page 12: Figure 4a: How can the large differences between the background colour and the local rainfall ratio value be explained? Is this just an effect of the weighting carried out in 3.1.2?

These differences can indeed be partly explained by the weighting carried out in 3.1.2, but also by the conservatism of the method, which tends to estimate a ratio closer to 1 when no significant signal can be detected.

Page 14: Figure 5 suggests that there is little difference between a) and c) and that, consequently, the correction provided by knowledge of b) is weak, whereas the structure of the image b) seems to show very high variability and a strong capacity to correct the raw image a).

-> Isn't this surprising? As a result, figure 5e) gives the impression that it is the WMA that contributes most to the correction of a)

-> Is this what we should understand?

Please note that the correction method was illustrated in Figure 5 over the Mont Blanc massif, where the ANTILOPE error is most significant. This can be seen in Figure 3 and Figure 7 (a black rectangle has been added to highlight the area covered by the precipitation fields of Figure 5), for a date with a pronounced artefact (no precipitation over the Mont Blanc summit and more than 20mm in nearby Chamonix). The precipitation fields used as an illustration in Figure 5 have been modified to avoid suggesting that the first correction step is ineffective.

The objective of the initial correction step aims at removing systematic biases estimated in the raw ANTILOPE product. As you point out, Figure 5c showed that this correction alone is not sufficient in extreme cases (in particular, if ANTILOPE fails to detect any precipitation, this multiplicative correction is indeed ineffective). Nevertheless, the impact of this correction on the estimated precipitation climatology is clearer in the new Figure 5. This is corroborated in Figure 7, which shows the yearly precipitation accumulation before and after the implementation of this first correction step alone (a black rectangle has been added to highlight the area covered by the precipitation fields of Figure 5). The mitigation of spatial artefacts over the Mont Blanc is notable, and systematic biases observed at reference stations are often reduced as expected.

You are right to point out that overall the WMA correction has a predominant impact on the structure final daily precipitation fields. However, it does not affect the climatology of the final product. This step is essential for the complete removal of spatial artefacts, but it produces daily precipitation fields that appear too smooth to be realistic on their own. These fields must therefore be interpreted in conjunction with the associated error fields that account for the removed spatial variability.

Page 14, this figure 5 could be enlarged to make it easier to read the axes and colorbars of the individual figures.

Figure 5 has been enlarged as suggested.

Line 460: The symbol "?" might be replaced by a reference or deleted

The reference to Germann et al., 2022 in line 460 (now 465) has been fixed.